# Solving a Class of Non-Convex Min-Max Games Using Iterative First Order Methods

**Maher Nouiehed**
nouiehed@usc.edu [*]

**Maziar Sanjabi**
sanjabi@usc.edu [†]

**Tianjian Huang**
tianjian@usc.edu [‡]

**Jason D. Lee**
jasonlee@princeton.edu [§]

**Meisam Razaviyayn**
razaviya@usc.edu [¶]

## Abstract

Recent applications that arise in machine learning have surged significant interest in solving min-max saddle point games. This problem has been extensively studied in the convex-concave regime for which a global equilibrium solution can be computed efficiently. In this paper, we study the problem in the non-convex regime and show that an $\varepsilon$–first order stationary point of the game can be computed when one of the player's objective can be optimized to global optimality efficiently. In particular, we first consider the case where the objective of one of the players satisfies the Polyak-Łojasiewicz (PL) condition. For such a game, we show that a simple multi-step gradient descent-ascent algorithm finds an $\varepsilon$–first order stationary point of the problem in $\widetilde{\mathcal{O}}(\varepsilon^{-2})$ iterations. Then we show that our framework can also be applied to the case where the objective of the "max-player" is concave. In this case, we propose a multi-step gradient descent-ascent algorithm that finds an $\varepsilon$–first order stationary point of the game in $\widetilde{\mathcal{O}}(\varepsilon^{-3.5})$ iterations, which is the best known rate in the literature. We applied our algorithm to a fair classification problem of Fashion-MNIST dataset and observed that the proposed algorithm results in smoother training and better generalization.

## 1  Introduction

Recent years have witnessed a wide range of machine learning and robust optimization applications being formulated as a min-max saddle point game; see [51, 11, 10, 50, 20, 53] and the references therein. Examples of problems that are formulated under this framework include generative adversarial networks (GANs) [51], reinforcement learning [11], adversarial learning [53], learning exponential families [10], fair statistical inference [17, 56, 52, 37], generative adversarial imitation learning [6, 27], distributed non-convex optimization [35] and many others. These applications require solving an optimization problem of the form

$$\min_{\boldsymbol{\theta} \in \Theta} \max_{\boldsymbol{\alpha} \in \mathcal{A}} \quad f(\boldsymbol{\theta}, \boldsymbol{\alpha}). \tag{1}$$

This optimization problem can be viewed as a zero-sum game between two players. The goal of the first player is to minimize $f(\boldsymbol{\theta}, \boldsymbol{\alpha})$ by tuning $\boldsymbol{\theta}$, while the other player's objective is to maximize

---

[*]Department of Industrial and Systems Engineering, University of Southern California

[†]Data Science and Operations Department, Marshall School of Business, University of Southern California

[‡]Department of Industrial and Systems Engineering, University of Southern California

[§]Department of Electrical Engineering, Princeton University

[¶]Department of Industrial and Systems Engineering, University of Southern California

$f(\boldsymbol{\theta}, \boldsymbol{\alpha})$ by tuning $\boldsymbol{\alpha}$. Gradient-based methods, especially gradient descent-ascent (GDA), are widely used in practice to solve these problems. GDA alternates between a gradient ascent steps on $\boldsymbol{\alpha}$ and a gradient descent steps on $\boldsymbol{\theta}$. Despite its popularity, this algorithm fails to converge even for simple bilinear zero-sum games [41, 39, 14, 2, 32]. This failure was fixed by adding negative momentum or by using primal-dual methods proposed by [22, 21, 8, 13, 15, 33].

When the objective $f$ is convex in $\boldsymbol{\theta}$ and concave in $\boldsymbol{\alpha}$, the corresponding variational inequality becomes monotone. This setting has been extensively studied and different algorithms have been developed for finding a Nash equilibrium [46, 21, 44, 29, 40, 23, 26, 43, 18, 45]. Moreover, [12] proposed an algorithm for solving a more general setting that covers both monotone and psuedo-monotone variational problems.

While the convex-concave setting has been extensively studied in the literature, recent machine learning applications urge the necessity of moving beyond these classical settings. For example, in a typical GAN problem formulation, two neural networks (generator and discriminator) compete in a non-convex zero-sum game framework [24]. For general non-convex non-concave games, [28, Proposition 10] provides an example for which local Nash equilibrium does not exist. Similarly, one can show that even second-order Nash equilibrium may not exist for non-convex games, see Section 2 for more details. Therefore, a well-justified objective is to find first order Nash equilibria of such games [48]; see definitions and discussion in Section 2. The first order Nash equilibrium can be viewed as a direct extension of the concept of first order stationarity in optimization to the above min-max game. While $\varepsilon$–first order stationarity in the context of optimization can be found efficiently in $\mathcal{O}(\varepsilon^{-2})$ iterations with gradient descent algorithm [47], the question of whether it is possible to design a gradient-based algorithm that can find an $\varepsilon$–first order Nash equilibrium for general non-convex saddle point games remains open.

Several recent results provided a partial answer to the problem of finding first-order stationary points of a non-convex min-max game. For instance, [51] proposed a stochastic gradient descent algorithm for the case when $f(\cdot, \cdot)$ is strongly concave in $\boldsymbol{\alpha}$ and showed convergence of the algorithm to an $\varepsilon$–first-order Nash equilibrium with $\widetilde{\mathcal{O}}(\varepsilon^{-2})$ gradient evaluations. Also, the work [28] analyzes the gradient descent algorithm with Max-oracle and shows $O(\epsilon^{-4})$ gradient evaluations and max-oracle calls for solving min-max problems where the inner problem can be solved in one iteration using an existing oracle. More recently, [35, 36] considered the case where $f$ is concave in $\boldsymbol{\alpha}$. They developed a descent-ascent algorithm with iteration complexity $\widetilde{\mathcal{O}}(\varepsilon^{-4})$. In this non-convex concave setting, [50] proposed a stochastic sub-gradient descent method with worst-case complexity $\widetilde{\mathcal{O}}(\varepsilon^{-6})$. Under the same concavity assumption on $f$, in this paper, we propose an alternative multi-step framework that finds an $\varepsilon$–first order Nash equilibrium/stationary with $\widetilde{\mathcal{O}}(\varepsilon^{-3.5})$ gradient evaluations.

In an effort to solve the more general non-convex non-concave setting, [34] developed a framework that converges to $\varepsilon$-first order stationarity/Nash equilibrium under the assumption that there exists a solution to the Minty variational inequality at each iteration. Although among the first algorithms with have theoretical convergence guarantees in the non-convex non-concave setting, the conditions required are strong and difficult to check. To the best of our knowledge, there is no practical problem for which the Minty variational inequality condition has been proven. With the motivation of exploring the non-convex non-concave setting, we propose a simple multi-step gradient descent ascent algorithm for the case where the objective of one of the players satisfies the Polyak-Łojasiewicz (PL) condition. We show the worst-case complexity of $\widetilde{\mathcal{O}}(\varepsilon^{-2})$ for our algorithm. This rate is optimal in terms of dependence on $\varepsilon$ up to logarithmic factors as discussed in Section 3. Compared to Minty variational inequality condition used in [34], the PL condition is very well studied in the literature and has been theoretically verified for objectives of optimization problems arising in many practical problems. For example, it has been proven to be true for objectives of over-parameterized deep networks [16], learning LQR models [19], phase retrieval [54], and many other simple problems discussed in [30]. In the context of min-max games, it has also been proven useful in generative adversarial imitation learning with LQR dynamics [6], as discussed in Section 3.

The rest of this paper is organized as follows. In Section 2 we define the concepts of First-order Nash equilibrium (FNE) and $\varepsilon$–FNE. In Section 3, we describe our algorithm designed for min-max games with the objective of one player satisfying the PL condition. Finally, in Section 4 we describe our method for solving games in which the function $f(\theta, \alpha)$ is concave in $\alpha$ (or convex in $\theta$).

## 2 Two-player Min-Max Games and First-Order Nash Equilibrium

Consider the two-player zero sum min-max game

$$\min_{\boldsymbol{\theta} \in \Theta} \max_{\boldsymbol{\alpha} \in \mathcal{A}} \ f(\boldsymbol{\theta}, \boldsymbol{\alpha}), \tag{2}$$

where $\Theta$ and $\mathcal{A}$ are both convex sets, and $f(\boldsymbol{\theta}, \boldsymbol{\alpha})$ is a continuously differentiable function. We say $(\boldsymbol{\theta}^*, \boldsymbol{\alpha}^*) \in \Theta \times \mathcal{A}$ is a *Nash equilibrium* of the game if

$$f(\boldsymbol{\theta}^*, \boldsymbol{\alpha}) \leq f(\boldsymbol{\theta}^*, \boldsymbol{\alpha}^*) \leq f(\boldsymbol{\theta}, \boldsymbol{\alpha}^*) \quad \forall \boldsymbol{\theta} \in \Theta, \ \forall \boldsymbol{\alpha} \in \mathcal{A}.$$

In convex-concave games, such a Nash equilibrium always exists [28] and several algorithms were proposed to find Nash equilibria [23, 26]. However, in the non-convex non-concave regime, computing these points is in general NP-Hard. In fact, even finding local Nash equilibria is NP-hard in the general non-convex non-concave regime. In addition, as shown by [28, Proposition 10], local Nash equilibria for general non-convex non-concave games may not exist. Thus, in this paper we aim for the less ambitious goal of finding *first-order Nash equilibrium* which is defined in the sequel.

**Definition 2.1** (FNE). *A point* $(\boldsymbol{\theta}^*, \boldsymbol{\alpha}^*) \in \Theta \times \mathcal{A}$ *is a first order Nash equilibrium (FNE) of the game* (2) *if*

$$\langle \nabla_{\boldsymbol{\theta}} f(\boldsymbol{\theta}^*, \boldsymbol{\alpha}^*), \boldsymbol{\theta} - \boldsymbol{\theta}^* \rangle \geq 0 \quad \forall \, \boldsymbol{\theta} \in \Theta \quad \text{and} \quad \langle \nabla_{\boldsymbol{\alpha}} f(\boldsymbol{\theta}^*, \boldsymbol{\alpha}^*), \boldsymbol{\alpha} - \boldsymbol{\alpha}^* \rangle \leq 0 \quad \forall \, \boldsymbol{\alpha} \in \mathcal{A}. \tag{3}$$

Notice that this definition, which is also used in [48, 49], contains the first order necessary optimality conditions of the objective function of each player [5]. Thus they are necessary conditions for local Nash equilibrium. Moreover, in the absence of constraints, the above definition simplifies to $\nabla_{\boldsymbol{\theta}} f(\boldsymbol{\theta}^*, \boldsymbol{\alpha}^*) = 0$ and $\nabla_{\boldsymbol{\alpha}} f(\boldsymbol{\theta}^*, \boldsymbol{\alpha}^*) = 0$, which are the well-known unconstrained first-order optimality conditions. Based on this observation, it is tempting to think that the above first-order Nash equilibrium condition does not differentiate between the min-max type solutions of (2) and min-min solutions of the type $\min_{\boldsymbol{\theta} \in \Theta, \boldsymbol{\alpha} \in \mathcal{A}} f(\boldsymbol{\theta}, \boldsymbol{\alpha})$. However, the direction of the second inequality in (3) would be different if we have considered the min-min problem instead of min-max problem. This different direction makes the problem of finding a FNE non-trivial. The following theorem guarantees the existence of first-order Nash equilibria under some mild assumptions.

**Theorem 2.2** (Restated from Proposition 2 in [48]). *Suppose the sets* $\Theta$ *and* $\mathcal{A}$ *are no-empty, compact, and convex. Moreover, assume that the function* $f(\cdot, \cdot)$ *is twice continuously differentiable. Then there exists a feasible point* $(\bar{\boldsymbol{\theta}}, \bar{\boldsymbol{\alpha}})$ *that is first-order Nash equilibrium.*

The above theorem guarantees existence of FNE points even when (local) Nash equilibria may not exist. The next natural question is about the computability of such methods. Since in practice we use iterative methods for computation, we need to define the notion of approximate–FNE.

**Definition 2.3** (Approximate FNE). *A point* $(\boldsymbol{\theta}^*, \boldsymbol{\alpha}^*)$ *is said to be an* $\varepsilon$*–first-order Nash equilibrium* ($\varepsilon$*–FNE) of the game* (2) *if*

$$\mathcal{X}(\boldsymbol{\theta}^*, \boldsymbol{\alpha}^*) \leq \varepsilon \quad \text{and} \quad \mathcal{Y}(\boldsymbol{\theta}^*, \boldsymbol{\alpha}^*) \leq \varepsilon,$$

*where*

$$\mathcal{X}(\boldsymbol{\theta}^*, \boldsymbol{\alpha}^*) \triangleq - \min_{\boldsymbol{\theta}} \ \langle \nabla_{\boldsymbol{\theta}} f(\boldsymbol{\theta}^*, \boldsymbol{\alpha}^*), \boldsymbol{\theta} - \boldsymbol{\theta}^* \rangle \ \text{s.t.} \ \boldsymbol{\theta} \in \Theta, \|\boldsymbol{\theta} - \boldsymbol{\theta}^*\| \leq 1, \tag{4}$$

*and*

$$\mathcal{Y}(\boldsymbol{\theta}^*, \boldsymbol{\alpha}^*) \triangleq \max_{\boldsymbol{\alpha}} \ \langle \nabla_{\boldsymbol{\alpha}} f(\boldsymbol{\theta}, \boldsymbol{\alpha}), \boldsymbol{\alpha} - \boldsymbol{\alpha}^* \rangle \ \text{s.t.} \ \boldsymbol{\alpha} \in \mathcal{A}, \|\boldsymbol{\alpha} - \boldsymbol{\alpha}^*\| \leq 1. \tag{5}$$

In the absence of constraints, $\varepsilon$–FNE in Definition 2.3 reduces to $\|\nabla_{\boldsymbol{\theta}} f(\boldsymbol{\theta}^*, \boldsymbol{\alpha}^*)\| \leq \varepsilon$ and $\|\nabla_{\boldsymbol{\alpha}} f(\boldsymbol{\theta}^*, \boldsymbol{\alpha}^*)\| \leq \varepsilon$.

**Remark 2.4.** *The* $\varepsilon$*–FNE definition above is based on the first order optimality measure of the objective of each player. Such first-order optimality measure has been used before in the context of optimization; see [9]. Such a condition guarantees that each player cannot improve their objective function using first order information. Similar to the optimization setting, one can define the second-order Nash equilibrium as a point that each player cannot improve their objective further by using first and second order information of their objectives. However, the use of second order Nash equilibria is more subtle in the context of games. The following example shows that such a point may not exist. Consider the game*

$$\min_{-1 \leq \theta \leq 1} \ \max_{-2 \leq \alpha \leq 2} \ -\theta^2 + \alpha^2 + 4\theta\alpha.$$

*Then* $(0, 0)$ *is the only first-order Nash equilibrium and is not a second-order Nash equilibrium.*

In this paper, our goal is to find an $\varepsilon$–FNE of the game (2) using iterative methods. To proceed, we make the following standard assumptions about the smoothness of the objective function $f$.

**Assumption 2.5.** *The function $f$ is continuously differentiable in both $\boldsymbol{\theta}$ and $\boldsymbol{\alpha}$ and there exists constants $L_{11}$, $L_{22}$ and $L_{12}$ such that for every $\boldsymbol{\alpha}, \boldsymbol{\alpha}_1, \boldsymbol{\alpha}_2 \in \mathcal{A}$, and $\boldsymbol{\theta}, \boldsymbol{\theta}_1, \boldsymbol{\theta}_2 \in \Theta$, we have*

$$\|\nabla_{\boldsymbol{\theta}} f(\boldsymbol{\theta}_1, \boldsymbol{\alpha}) - \nabla_{\boldsymbol{\theta}} f(\boldsymbol{\theta}_2, \boldsymbol{\alpha})\| \le L_{11}\|\boldsymbol{\theta}_1 - \boldsymbol{\theta}_2\|, \quad \|\nabla_{\boldsymbol{\alpha}} f(\boldsymbol{\theta}, \boldsymbol{\alpha}_1) - \nabla_{\boldsymbol{\alpha}} f(\boldsymbol{\theta}, \boldsymbol{\alpha}_2)\| \le L_{22}\|\boldsymbol{\alpha}_1 - \boldsymbol{\alpha}_2\|,$$
$$\|\nabla_{\boldsymbol{\alpha}} f(\boldsymbol{\theta}_1, \boldsymbol{\alpha}) - \nabla_{\boldsymbol{\alpha}} f(\boldsymbol{\theta}_2, \boldsymbol{\alpha})\| \le L_{12}\|\boldsymbol{\theta}_1 - \boldsymbol{\theta}_2\|, \quad \|\nabla_{\boldsymbol{\theta}} f(\boldsymbol{\theta}, \boldsymbol{\alpha}_1) - \nabla_{\boldsymbol{\theta}} f(\boldsymbol{\theta}, \boldsymbol{\alpha}_2)\| \le L_{12}\|\boldsymbol{\alpha}_1 - \boldsymbol{\alpha}_2\|.$$

# 3  Non-Convex PL-Game

In this section, we consider the problem of developing an "efficient" algorithm for finding an $\varepsilon$–FNE of (2) when the objective of one of the players satistys Polyak-Łojasiewicz (PL) condition. To proceed, let us first formally define the Polyak-Łojasiewicz (PL) condition.

**Definition 3.1** (Polyak-Łojasiewicz Condition)**.** *A differentiable function $h(\mathbf{x})$ with the minimum value $h^* = \min_x h(\mathbf{x})$ is said to be $\mu$-Polyak-Łojasiewicz ($\mu$-PL) if*

$$\frac{1}{2}\|\nabla h(\mathbf{x})\|^2 \ge \mu(h(\mathbf{x}) - h^*), \quad \forall x. \tag{6}$$

The PL-condition has been established and utilized for analyzing many practical modern problems [30, 19, 16, 54, 6]. Moreover, it is well-known that a function can be non-convex and still satisfy the PL condition [30]. Based on the definition above, we define a class of min-max PL-games.

**Definition 3.2** (PL-Game)**.** *We say that the min-max game (2) is a PL-Game if the max player is unconstrained, i.e., $\mathcal{A} = \mathbb{R}^n$, and there exists a constant $\mu > 0$ such that the function $h_{\boldsymbol{\theta}}(\boldsymbol{\alpha}) \triangleq -f(\boldsymbol{\theta}, \boldsymbol{\alpha})$ is $\mu$-PL for any fixed value of $\boldsymbol{\theta} \in \Theta$.*

A simple example of a practical PL-game is detailed next.

**Example 3.1** (Generative adversarial imitation learning of linear quadratic regulators)**.** *Imitation learning is a paradigm that aims to learn from an expert's demonstration of performing a task [6]. It is known that this learning process can be formulated as a min-max game [27]. In such a game the minimization is performed over all the policies and the goal is to minimize the discrepancy between the accumulated reward for expert's policy and the proposed policy. On the other hand, the maximization is done over the parameters of the reward function and aims at maximizing this discrepancy over the parameters of the reward function. This approach is also referred to as generative adversarial imitation learning (GAIL) [27]. The problem of generative adversarial imitation learning for linear quadratic regulators [6] refers to solving this problem for the specific case where the underlying dynamic and the reward function come from a linear quadratic regulator [19]. To be more specific, this problem can be formulated [6] as $\min_K \max_{\boldsymbol{\theta} \in \Theta} m(K, \boldsymbol{\theta})$, where $K$ represents the choice of the policy and $\boldsymbol{\theta}$ represents the parameters of the dynamic and the reward functions. Under the discussed setting, $m$ is strongly concave in $\boldsymbol{\theta}$ and PL in $K$ (see [6] for more details). Note that since $m$ is strongly concave in $\boldsymbol{\theta}$ and $PL$ in $K$, any FNE of the game would also be a Nash equilibrium point. Also note that the notion of FNE does not depend on the ordering of the $\min$ and $\max$. Thus, to be consistent with our notion of PL-games, we can formulate the problem as*

$$\min_{\boldsymbol{\theta} \in \Theta} \max_{K} -m(K, \boldsymbol{\theta}) \tag{7}$$

*Thus, generative adversarial imitation learning of linear quadratic regulators is an example of finding a FNE for a min-max PL-game.*

In what follows, we present a simple iterative method for computing an $\varepsilon$–FNE of PL games.

## 3.1  Multi-step gradient descent ascent for PL-games

In this section, we propose a multi-step gradient descent ascent algorithm that finds an $\varepsilon$–FNE point for PL-games. At each iteration, our method runs multiple projected gradient ascent steps to estimate the solution of the inner maximization problem. This solution is then used to estimate the gradient of the inner maximization value function, which directly provides a descent direction. In a nutshell, our proposed algorithm is a gradient descent-like algorithm on the inner maximization value function. To present the ideas of our multi-step algorithm, let us re-write (2) as

$$\min_{\boldsymbol{\theta} \in \Theta} g(\boldsymbol{\theta}), \tag{8}$$

where

$$g(\boldsymbol{\theta}) \triangleq \max_{\boldsymbol{\alpha} \in \mathcal{A}} f(\boldsymbol{\theta}, \boldsymbol{\alpha}). \tag{9}$$

A famous classical result in optimization is Danskin's theorem [4] which provides a sufficient condition under which the gradient of the value function $\max_{\boldsymbol{\alpha} \in \mathcal{A}} f(\boldsymbol{\theta}, \boldsymbol{\alpha})$ can be directly evaluated using the gradient of the objective $f(\boldsymbol{\theta}, \boldsymbol{\alpha}^*)$ evaluated at the optimal solution $\boldsymbol{\alpha}^*$. This result requires the optimizer $\boldsymbol{\alpha}^*$ to be unique. Under our PL assumption on $f(\boldsymbol{\theta}, \cdot)$, the inner maximization problem (9) may have multiple optimal solutions. Hence, Danskin's theorem does not directly apply. However, as we will show in Lemma A.5 in the supplementary, under the PL assumption, we still can show the following result

$$\nabla_{\boldsymbol{\theta}} g(\boldsymbol{\theta}) = \nabla_{\boldsymbol{\theta}} f(\boldsymbol{\theta}, \boldsymbol{\alpha}^*) \quad \text{with} \quad \boldsymbol{\alpha}^* \in \arg\max_{\boldsymbol{\alpha} \in \mathcal{A}} f(\boldsymbol{\theta}, \boldsymbol{\alpha}),$$

despite the non-uniqueness of the optimal solution.

Motivated by this result, we propose a Multi-step Gradient Descent Ascent algorithm that solves the inner maximization problem to "approximate" the gradient of the value function $g$. This gradient direction is then used to descent on $\boldsymbol{\theta}$. More specifically, the inner loop (Step 4) in Algorithm 1 solves the maximization problem (9) for a given fixed value $\boldsymbol{\theta} = \boldsymbol{\theta}_t$. The computed solution of this optimization problem provides an approximation for the gradient of the function $g(\boldsymbol{\theta})$, see Lemma A.6 in Appendix A. This gradient is then used in Step 7 to descent on $\boldsymbol{\theta}$.

---

**Algorithm 1** Multi-step Gradient Descent Ascent

1: INPUT: $K, T, \eta_1 = 1/L_{22}, \eta_2 = 1/L, \boldsymbol{\alpha}_0 \in \mathcal{A}$ and $\boldsymbol{\theta}_0 \in \Theta$
2: **for** $t = 0, \cdots, T-1$ **do**
3:      Set $\boldsymbol{\alpha}_0(\boldsymbol{\theta}_t) = \boldsymbol{\alpha}_t$
4:      **for** $k = 0, \cdots, K-1$ **do**
5:          Set $\boldsymbol{\alpha}_{k+1}(\boldsymbol{\theta}_t) = \boldsymbol{\alpha}_k(\boldsymbol{\theta}_t) + \eta_1 \nabla_{\boldsymbol{\alpha}} f(\boldsymbol{\theta}_t, \boldsymbol{\alpha}_k(\boldsymbol{\theta}_t))$
6:      **end for**
7:      Set $\boldsymbol{\theta}_{t+1} = \text{proj}_\Theta \left( \boldsymbol{\theta}_t - \eta_2 \nabla_{\boldsymbol{\theta}} \text{f}(\boldsymbol{\theta}_t, \boldsymbol{\alpha}_K(\boldsymbol{\theta}_t)) \right)$
8: **end for**
9: Return $(\boldsymbol{\theta}_t, \boldsymbol{\alpha}_K(\boldsymbol{\theta}_t))$ for $t = 0, \cdots, T-1$.

---

### 3.2    Convergence analysis of Multi-Step Gradient Descent Ascent Algorithm for PL games

Throughout this section, we make the following assumption.

**Assumption 3.3.** *The constraint set $\Theta$ is convex and compact. Moreover, there exists a ball with radius $R$, denoted by $\mathcal{B}_R$, such that $\Theta \subseteq \mathcal{B}_R$.*

We are now ready to state the main result of this section.

**Theorem 3.4.** *Under Assumptions 2.5 and 3.3, for any given scalar $\varepsilon \in (0, 1)$, if we choose $K$ and $T$ large enough such that*

$$T \geq N_T(\varepsilon) \triangleq \mathcal{O}(\varepsilon^{-2}) \quad \text{and} \quad K \geq N_K(\varepsilon) \triangleq \mathcal{O}(\log(\varepsilon^{-1})),$$

*then there exists an iteration $t \in \{0, \cdots, T\}$ such that $(\boldsymbol{\theta}_t, \boldsymbol{\alpha}_{t+1})$ is an $\varepsilon$–FNE of (2).*

*Proof.* The proof is relegated to Appendix A.2.         $\square$

**Corollary 3.5.** *Under Assumption 2.5 and Assumption 3.3, Algorithm 1 finds an $\varepsilon$-FNE of the game (2) with $\mathcal{O}(\varepsilon^{-2})$ gradient evaluations of the objective with respect to $\boldsymbol{\theta}$ and $\mathcal{O}(\varepsilon^{-2} \log(\varepsilon^{-1}))$ gradient evaluations with respect to $\boldsymbol{\alpha}$. If the two gradient oracles have the same complexity, the overall complexity of the method would be $\mathcal{O}(\varepsilon^{-2} \log(\varepsilon^{-1}))$.*

**Remark 3.6.** *The iteration complexity order $\mathcal{O}(\varepsilon^{-2} \log(\varepsilon^{-1}))$ in Theorem 3.4 is tight (up to logarithmic factors). This is due to the fact that for general non-convex smooth problems, finding an $\varepsilon$–stationary solution requires at least $\Omega(\varepsilon^{-2})$ gradient evaluations [7, 47]. Clearly, this lower bound is also valid for finding an $\varepsilon$–FNE of PL-games. This is because we can assume that the function $f(\boldsymbol{\theta}, \boldsymbol{\alpha})$ does not depend on $\boldsymbol{\alpha}$ (and thus PL in $\boldsymbol{\alpha}$).*

**Remark 3.7.** *Theorem 3.4 shows that under the PL assumption, the pair $(\boldsymbol{\theta}_t, \boldsymbol{\alpha}_K(\theta_t))$ computed by Algorithm 1 is an $\varepsilon$–FNE of the game* (2). *Since $\boldsymbol{\alpha}_K(\theta_t)$ is an approximate solution of the inner maximization problem, we get that $\boldsymbol{\theta}_t$ is concurrently an $\varepsilon$–first order stationary solution of the optimization problem* (8).

**Remark 3.8.** *In [51, Theorem 4.2], a similar result was shown for the case when $f(\boldsymbol{\theta}, \boldsymbol{\alpha})$ is strongly concave in $\boldsymbol{\alpha}$. Hence, Theorem 3.4 can be viewed as an extension of [51, Theorem 4.2]. Similar to [51, Theorem 4.2], one can easily extend the result of Theorem 3.4 to the stochastic setting by replacing the gradient of $f$ with respect to $\theta$ in Step 7 by the stochastic version of the gradient.*

In the next section we consider the non-convex concave min-max saddle game. It is well-known that convexity/concavity does not imply the PL condition and PL condition does not imply convexity/concavity [30]. Therefore, the problems we consider in the next section are neither restriction nor extension of our results on PL games.

## 4 Non-Convex Concave Games

In this section, we focus on "non-convex concave" games satisfying the following assumption:

**Assumption 4.1.** *The objective function $f(\boldsymbol{\theta}, \boldsymbol{\alpha})$ is concave in $\boldsymbol{\alpha}$ for any fixed value of $\boldsymbol{\theta}$. Moreover, the set $\mathcal{A}$ is convex and compact, and there exists a ball with radius $R$ that contains the feasible set $\mathcal{A}$.*

One major difference of this case with the PL-games is that in this case the function $g(\boldsymbol{\theta}) = \max_{\boldsymbol{\alpha} \in \mathcal{A}} f(\boldsymbol{\theta}, \boldsymbol{\alpha})$ might not be differentiable. To see this, consider the example $g(\alpha) = \max_{0 \le \alpha \le 1}(2\alpha - 1)\theta$ which is concave in $\alpha$. However, the value function $g(\theta) = |\theta|$ is non-smooth.

Using a small regularization term, we approximate the function $g(\cdot)$ by a differentiable function

$$g_\lambda(\boldsymbol{\theta}) \triangleq \max_{\boldsymbol{\alpha} \in \mathcal{A}} \ f_\lambda(\boldsymbol{\theta}, \boldsymbol{\alpha}), \tag{10}$$

where $f_\lambda(\boldsymbol{\theta}, \boldsymbol{\alpha}) \triangleq f(\boldsymbol{\theta}, \boldsymbol{\alpha}) - \frac{\lambda}{2}\|\boldsymbol{\alpha} - \bar{\boldsymbol{\alpha}}\|^2$. Here $\bar{\boldsymbol{\alpha}} \in \mathcal{A}$ is some given fixed point and $\lambda > 0$ is a regularization parameter that we will specify later. Since $f(\boldsymbol{\theta}, \boldsymbol{\alpha})$ is concave in $\boldsymbol{\alpha}$, $f_\lambda(\boldsymbol{\theta}, \cdot)$ is $\lambda$-strongly concave. Thus, the function $g_\lambda(\cdot)$ becomes smooth with Lipschitz gradient; see Lemma B.1 in the supplementary. Using this property, we propose an algorithm that runs at each iteration multiple steps of Nesterov accelerated projected gradient ascent to estimate the solution of (10). This solution is then used to estimate the gradient of $g_\lambda(\boldsymbol{\theta})$ which directly provides a descent direction on $\boldsymbol{\theta}$. Our algorithm computes an $\varepsilon$–FNE point for non-convex concave games with $\widetilde{\mathcal{O}}(\varepsilon^{-3.5})$ gradient evaluations. Then for sufficiently small regularization coefficient, we show that the computed point is an $\varepsilon$-FNE.

Notice that since $f_\lambda$ is Lipschitz smooth and based on the compactness assumption, we can define

$$g_{\boldsymbol{\theta}} \triangleq \max_{\boldsymbol{\theta} \in \Theta} \|\nabla g_\lambda(\boldsymbol{\theta})\|, \ \ g_{\boldsymbol{\alpha}} \triangleq \max_{\boldsymbol{\theta} \in \Theta} \|\nabla_{\boldsymbol{\alpha}} f_\lambda(\boldsymbol{\theta}, \boldsymbol{\alpha}^*(\boldsymbol{\theta}))\|, \ \text{ and } \ g_{max} = \max\{g_{\boldsymbol{\theta}}, g_{\boldsymbol{\alpha}}, 1\}, \tag{11}$$

where $\boldsymbol{\alpha}^*(\boldsymbol{\theta}) \triangleq \arg\max_{\boldsymbol{\alpha} \in \mathcal{A}} \ f_\lambda(\boldsymbol{\theta}, \boldsymbol{\alpha})$. We are now ready to describe our proposed algorithm.

### 4.1 Algorithm Description

Our proposed method is outlined in Algorithm 2. This algorithm has two steps: step 2 and step 3. In step 2, $K$ steps of accelerated gradient ascent method is run over the variable $\boldsymbol{\alpha}$ to find an approximate maximizer of the problem $\max_{\boldsymbol{\alpha}} \ f_\lambda(\boldsymbol{\theta}_t, \boldsymbol{\alpha})$. Then using approximate maximizer $\boldsymbol{\alpha}_{t+1}$, we update $\boldsymbol{\theta}$ variable using one step of first order methods in step 3.

In step 2, we run $K$ step of accelerated gradient ascent algorithm over the variable $\alpha$ with restart every $N$ iterations. The details of this subroutine can be found in subsection B.1 of the supplementary materials. In step 3 of Algorithm 2, we can either use projected gradient descent update rule

$$\boldsymbol{\theta}_{t+1} \triangleq \text{proj}_\Theta\left(\boldsymbol{\theta}_t - \frac{1}{L_{11} + L_{12}^2/\lambda}\nabla_{\boldsymbol{\theta}} f_\lambda(\boldsymbol{\theta}_t, \boldsymbol{\alpha}_{t+1})\right),$$

or Frank-Wolfe update rule described in subsection B.2 in the supplementary material. We show convergence of the algorithm to $\varepsilon$–FNE in Theorems 4.2.

**Algorithm 2** Multi-Step Frank Wolfe/Projected Gradient Step Framework

**Require:** Constants $\widetilde{L} \triangleq \max\{L, L_{12}, g_{max}\}$, $N \triangleq \lfloor \sqrt{8L_{22}/\lambda} \rfloor$, $K, T, \eta, \lambda, \boldsymbol{\theta}_0 \in \Theta, \boldsymbol{\alpha}_0 \in \mathcal{A}$

1: **for** $t = 0, 1, 2, \ldots, T$ **do**
2:     Set $\boldsymbol{\alpha}_{t+1} = \text{APGA}(\boldsymbol{\alpha}_t, \boldsymbol{\theta}_t, \eta, N, K)$ by running $K$ steps of Accelerated Projected Gradient Ascent subroutine (Algorithm 3) with periodic restart at every $N$ iteration.
3:     Compute $\boldsymbol{\theta}_{t+1}$ using first-order information (Frank-Wolfe or projected gradient descent).
4: **end for**

---

**Theorem 4.2.** *Given a scalar $\varepsilon \in (0, 1)$. Assume that Step 3 in Algorithm 2 sets either runs projected gradient descent or Frank-Wolfe iteration. Under Assumptions 4.1 and 2.5,*

$$\eta = \frac{1}{L_{22}}, \quad \lambda \triangleq \frac{\varepsilon}{4R}, \quad T \geq N_T(\varepsilon) \triangleq \mathcal{O}(\varepsilon^{-3}), \quad \text{and} \quad K \geq N_K(\varepsilon) \triangleq \mathcal{O}\big(\varepsilon^{-1/2} \log(\varepsilon^{-1})\big),$$

*then there exists $t \in \{0, \ldots, T\}$ such that $(\boldsymbol{\theta}_t, \boldsymbol{\alpha}_{t+1})$ is an $\varepsilon$–FNE of problem (2).*

*Proof.* The proof is relegated to Appendix B.4. $\qquad\square$

**Corollary 4.3.** *Under Assumptions 2.5 and 4.1, Algorithm 2 finds an $\varepsilon$-first-order stationary solution of the game (2) with $\mathcal{O}(\varepsilon^{-3})$ gradient evaluations of the objective with respect $\boldsymbol{\theta}$ and $\mathcal{O}(\varepsilon^{-0.5} \log(\varepsilon^{-1}))$ gradient evaluations with respect to $\boldsymbol{\alpha}$. If the two oracles have the same complexity, the overall complexity of the method would be $\mathcal{O}(\varepsilon^{-3.5} \log(\varepsilon^{-1}))$.*

# 5 Numerical Results

We evaluate the numerical performance of Algorithm 2 in the following two applications:

## 5.1 Fair Classifier

We conduct two experiment on the Fashion MNIST dataset [55]. This dataset consists of $28 \times 28$ arrays of grayscale pixel images classified into 10 categories of clothing. It includes $60,000$ training images and $10,000$ testing images.

**Experimental Setup:** The recent work in [42] observed that training a logisitic regression model to classify the images of the Fashion MNIST dataset can be biased against certain categories. To remove this bias, [42] proposed to minimize the maximum loss incurred by the different categories. We repeat the experiment when using a more complex non-convex Convolutional Neural Network (CNN) model for classification. Similar to [42], we limit our experiment to the three categories T-shirt/top, Coat, and Shirts, that correspond to the lowest three testing accuracies achieved by the trained classifier. To minimize the maximum loss over these three categories, we train the classifier to minimize

$$\min_{\mathbf{W}} \ \max \{\mathcal{L}_1(\mathbf{W}), \mathcal{L}_2(\mathbf{W}), \mathcal{L}_3(\mathbf{W})\}, \tag{12}$$

where $\mathbf{W}$ represents the parameters of the CNN; and $\mathcal{L}_1$, $\mathcal{L}_2$, and $\mathcal{L}_3$ correspond to the loss incurred by samples in T-shirt/top, Coat, and Shirt categories. Problem (12) can be re-written as

$$\min_{\mathbf{W}} \ \max_{t_1,t_2,t_3} \ \sum_{i=1}^{3} t_i \mathcal{L}_i(\mathbf{W}) \quad \text{s.t.} \quad t_i \geq 0 \quad \forall i = 1, 2, 3; \quad \sum_{i=1}^{3} t_i = 1.$$

Clearly the inner maximization problem is concave; and thus our theory can be applied. To empirically evaluate the regularization scheme proposed in Section 4, we implement two versions of Algorithm 2. The first version solves at each iteration the regularized strongly concave sub-problem

$$\max_{t_1,t_2,t_3} \ \sum_{i=1}^{3} t_i \mathcal{L}_i(\mathbf{W}) - \frac{\lambda}{2} \sum_{i=1}^{3} t_i^2 \quad \text{s.t.} \quad t_i \geq 0 \quad \forall i = 1, 2, 3; \quad \sum_{i=1}^{3} t_i = 1, \tag{13}$$

and use the optimum $t$ to perform a gradient descent step on $\mathbf{W}$ (notice that fixing the value of $\mathbf{W}$, the optimum $t$ can be computed using KKT conditions and a simple sorting or bisection procedure).

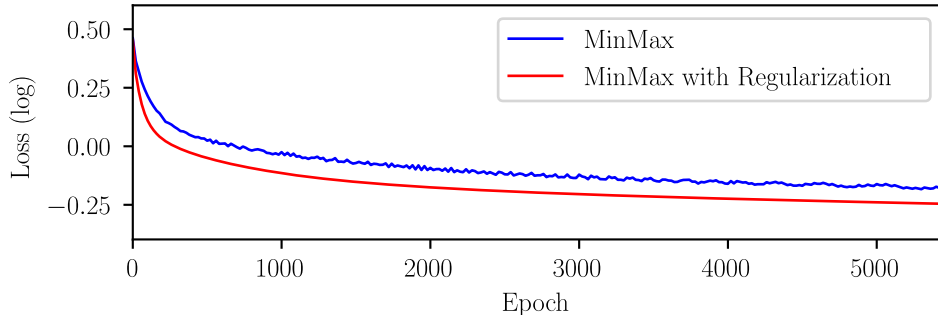

Figure 1: The effect of regularization on the convergence of the training loss, $\lambda = 0.1$.

The second version of Algorithm 2 solves at each iteration the concave inner maximization problem without the regularization term. Then uses the computed solution to perform a descent step on $\mathbf{W}$. Notice that in both cases, the optimization with respect to $t$ variable can be done in (almost) closed-form update. Although regularization is required to have theoretical convergence guarantees, we compare the two versions of the algorithm on empirical data to determine whether we lose by adding such regularization. We further compare these two algorithms with normal training that uses gradient descent to minimize the average loss among the three categories. We run all algorithms for 5500 epochs and record the test accuracy of the categories. To reduce the effect of random initialization, we run our methods with 50 different random initializations and record the average and standard deviation of the test accuracy collected. For fair comparison, the same initialization is used for all methods in each run. The results are summarized in Tables 1. To test our framework in stochastic settings, we repeat the experiment running all algorithms for 12,000 iterations with Adam and SGD optimizer with a bath size of 600 images (200 from each category). The results of the second experiment with Adam optimizer are summarized in Table 2. The model architecture and parameters are detailed in Appendix F. The choice of Adam optimizer is mainly because it is more robust to the choice of the step-size and thus can be easily tuned. In fact, the use of SGD or Adam does not change the overall takeaways of the experiments. The results of using SGD optimizer are relegated to Appendix C.

**Results:** Tables 1 and 2 show the average and standard deviation of the number of correctly classified samples. The average and standard deviation are taken over 50 runs. For each run 1000 testing samples are considered for each category. The results show that when using MinMax and MinMax with regularization, the accuracies across the different categories are more balanced compared to normal training. Moreover, the tables show that Algorithm 2 with regularization provides a slightly better worst-case performance compared to the unregularized approach. Note that the empirical advantages due to regularization appears more in the stochastic setting. To see this compare the differences between MinMax and MinMax with Regularization in Tables 1 and 2. Figure 1 depicts a sample trajectory of deterministic algorithm applied to the regularized and regularized formulations. This figures shows that regularization provides a smoother and slightly faster convergence compared to the unregularized approach. In addition, we apply our algorithm to the exact similar logistic regression setup as in [42]. Results of this experiment can be found in Appendix D.

| | T-shirt/top | | Coat | | Shirt | | Worst | |
| --- | --- | --- | --- | --- | --- | --- | --- | --- |
| | mean | std | mean | std | mean | std | mean | std |
| Normal | 850.72 | 8.58 | 843.50 | 17.24 | 658.74 | 17.81 | 658.74 | 17.81 |
| MinMax | 774.14 | 10.40 | 753.88 | 22.52 | 766.14 | 13.59 | 750.04 | 18.92 |
| MinMax with Regularization | 779.84 | 10.53 | 765.56 | 22.28 | 762.34 | 11.91 | **755.66** | **15.11** |

Table 1: The mean and standard deviation of the number of correctly classified samples when gradient descent is used in training, $\lambda = 0.1$.

| | T-shirt/top | | Coat | | Shirt | | Worst | |
|---|---|---|---|---|---|---|---|---|
| | mean | std | mean | std | mean | std | mean | std |
| Normal | 853.86 | 10.04 | 852.22 | 18.27 | 683.32 | 17.96 | 683.32 | 17.96 |
| MinMax | 753.44 | 15.12 | 715.24 | 32.00 | 733.42 | 18.51 | 711.64 | 29.02 |
| MinMax with Regularization | 764.02 | 14.12 | 739.80 | 27.60 | 748.84 | 15.79 | **734.34** | **23.54** |

Table 2: The mean and standard deviation of the number of correctly classified samples when Adam (mini-batch) is used in training, $\lambda = 0.1$.

## 5.2 Robust Neural Network Training

**Experimental Setup:** Neural networks have been widely used in various applications, especially in the field of image recognition. However, these neural networks are vulnerable to adversarial attacks, such as Fast Gradient Sign Method (FGSM) [25] and Projected Gradient Descent (PGD) attack [31]. These adversarial attacks show that a small perturbation in the data input can significantly change the output of a neural network. To train a robust neural network against adversarial attacks, researchers reformulate the training procedure into a robust min-max optimization formulation [38], such as

$$\min_{\mathbf{w}} \sum_{i=1}^{N} \max_{\delta_i, \text{ s.t. } |\delta_i|_{\infty} \leq \varepsilon} \ell(f(x_i + \delta_i; \mathbf{w}), y_i).$$

Here $\mathbf{w}$ is the parameter of the neural network, the pair $(x_i, y_i)$ denotes the $i$-th data point, and $\delta_i$ is the perturbation added to data point $i$. As discussed in this paper, solving such a non-convex non-concave min-max optimization problem is computationally challenging. Motivated by the theory developed in this work, we approximate the above optimization problem with a novel objective function which is concave in the parameters of the (inner) maximization player. To do so, we first approximate the inner maximization problem with a finite max problem

$$\min_{\mathbf{w}} \sum_{i=1}^{N} \max \left\{ \ell(f(\hat{x}_{i0}(\mathbf{w}); \mathbf{w}), y_i), \ldots, \ell(f(\hat{x}_{i9}(\mathbf{w}); \mathbf{w}), y_i) \right\}, \tag{14}$$

where each $\hat{x}_{ij}(\mathbf{w})$ is the result of a targeted attack on sample $x_i$ aiming at changing the output of the network to label $j$. These perturbed inputs, which are explained in details in Appendix E, are the function of the weights of the network. Then we replace this finite max inner problem with a concave problem over a probability simplex. Such a concave inner problem allows us to use the multi-step gradient descent-ascent method. The structure of the network and the details of the formulation is detailed in Appendix E.

**Results:** We compare our results with [38, 57]. Note [57] is the state-of-the-art algorithm and has won the first place, out of $\approx 2000$ submissions, in the NeurIPS 2018 Adversarial Vision Challenge. The accuracy of our formulation against popular attacks, FGSM [25] and PGD [31], are summarized in Table 3. This table shows that our formulation leads to a comparable results against state-of-the-art algorithms (while in some cases it also outperform those methods by as much as $\approx 15\%$ accuracy).

| | Natural | FGSM $L_{\infty}$ [25] | | | PGD$^{40}$ $L_{\infty}$ [31] | | |
|---|---|---|---|---|---|---|---|
| | | $\varepsilon = 0.2$ | $\varepsilon = 0.3$ | $\varepsilon = 0.4$ | $\varepsilon = 0.2$ | $\varepsilon = 0.3$ | $\varepsilon = 0.4$ |
| [38] with $\varepsilon = 0.35$ | **98.58%** | 96.09% | 94.82% | 89.84% | 94.64% | 91.41% | 78.67% |
| [57] with $\varepsilon = 0.35$ | 97.37% | 95.47% | 94.86% | 79.04% | 94.41% | 92.69% | 85.74% |
| [57] with $\varepsilon = 0.40$ | 97.21% | 96.19% | 96.17% | 96.14% | 95.01% | 94.36% | 94.11% |
| Proposed with $\varepsilon = 0.40$ | 98.20% | **97.04%** | **96.66%** | **96.23%** | **96.00%** | **95.17%** | **94.22%** |

Table 3: Test accuracies under FGSM and PGD attacks. All adversarial images are quantified to 256 levels ($0 - 255$ integer).

Links to code and pre-trained models of above two simulations are available at Appendix G.

