[Supplementary Material · Min_Max_Concave_PL_NIPS_appendix.pdf]

# A Proofs for results in Section 3

Before proceeding to the proofs of the main results, we need some intermediate lemmas and preliminary definitions.

**Definition A.1.** *[1] A function $h(\mathbf{x})$ is said to satisfy the Quadratic Growth (QG) condition with constant $\gamma > 0$ if*

$$h(\mathbf{x}) - h^* \geq \frac{\gamma}{2}\mathrm{dist}(\mathbf{x})^2, \quad \forall x,$$

*where $h^*$ is the minimum value of the function, and $\mathrm{dist}(\mathbf{x})$ is the distance of the point $x$ to the optimal solution set.*

The following lemma shows that PL implies QG [30].

**Lemma A.2** (Corollary of Theorem 2 in [30]). *If function $f$ is PL with constant $\mu$, then $f$ satisfies the quadratic growth condition with constant $\gamma = 4\mu$.*

The next Lemma shows the stability of $\arg\max_{\boldsymbol{\alpha}} f(\boldsymbol{\theta}, \boldsymbol{\alpha})$ with respect to $\boldsymbol{\theta}$ under PL condition.

**Lemma A.3.** *Assume that $\{h_{\boldsymbol{\theta}}(\boldsymbol{\alpha}) = -f(\boldsymbol{\theta}, \boldsymbol{\alpha}) \mid \boldsymbol{\theta}\}$ is a class of $\mu$-PL functions in $\boldsymbol{\alpha}$. Define $A(\boldsymbol{\theta}) = \arg\max_{\boldsymbol{\alpha}} f(\boldsymbol{\theta}, \boldsymbol{\alpha})$ and assume $A(\boldsymbol{\theta})$ is closed. Then for any $\boldsymbol{\theta}_1$, $\boldsymbol{\theta}_2$ and $\boldsymbol{\alpha}_1 \in A(\boldsymbol{\theta}_1)$, there exists an $\boldsymbol{\alpha}_2 \in A(\boldsymbol{\theta}_2)$ such that*

$$\|\boldsymbol{\alpha}_1 - \boldsymbol{\alpha}_2\| \leq \frac{L_{12}}{2\mu}\|\boldsymbol{\theta}_1 - \boldsymbol{\theta}_2\| \tag{15}$$

*Proof.* Based on the Lipchitzness of the gradients, we have that $\|\nabla_{\boldsymbol{\alpha}} f(\boldsymbol{\theta}_2, \boldsymbol{\alpha}_1)\| \leq L_{12}\|\boldsymbol{\theta}_1 - \boldsymbol{\theta}_2\|$. Then using the PL condition, we know that

$$g(\boldsymbol{\theta}_2) + h_{\boldsymbol{\theta}_2}(\boldsymbol{\alpha}_1) \leq \frac{L_{12}^2}{2\mu}\|\boldsymbol{\theta}_1 - \boldsymbol{\theta}_2\|^2. \tag{16}$$

Now we use the result of Lemma A.2 to show that there exists $\boldsymbol{\alpha}_2 = \arg\min_{\boldsymbol{\alpha}\in A(\boldsymbol{\theta}_2)} \|\boldsymbol{\alpha} - \boldsymbol{\alpha}_1\|^2 \in A(\boldsymbol{\theta}_2)$ such that

$$2\mu\|\boldsymbol{\alpha}_1 - \boldsymbol{\alpha}_2\|^2 \leq \frac{L_{12}^2}{2\mu}\|\boldsymbol{\theta}_1 - \boldsymbol{\theta}_2\|^2 \tag{17}$$

re-arranging the terms, we get the desired result that

$$\|\boldsymbol{\alpha}_1 - \boldsymbol{\alpha}_2\| \leq \frac{L_{12}}{2\mu}\|\boldsymbol{\theta}_1 - \boldsymbol{\theta}_2\|.$$

$\square$

Finally, the following lemma would be useful in the proof of Theorem 3.4.

**Lemma A.4** (See Theorem 5 in [30]). *Assume $h(\mathbf{x})$ is $\mu$-PL and $L$-smooth. Then, by applying gradient descent with step-size $1/L$ from point $x_0$ for $K$ iterations we get an $x_K$ such that*

$$h(\mathbf{x}) - h^* \leq \left(1 - \frac{\mu}{L}\right)^K (h(x_0) - h^*), \tag{18}$$

*where $h^* = \min_x h(\mathbf{x})$.*

We are now ready to prove the results in Section 3.

## A.1 Danskin-type Lemma for PL Functions

**Lemma A.5.** *Under Assumption 2.5 and PL-game assumption,*

$$\nabla_{\boldsymbol{\theta}} g(\boldsymbol{\theta}) = \nabla_{\boldsymbol{\theta}} f(\boldsymbol{\theta}, \boldsymbol{\alpha}^*), \quad where \quad \boldsymbol{\alpha}^* \in \arg\max_{\boldsymbol{\alpha}\in\mathcal{A}} f(\boldsymbol{\theta}, \boldsymbol{\alpha}).$$

*Moreover, $g$ is $L$-Lipschitz smooth with $L = L_{11} + \frac{L_{12}^2}{2\mu}$.*

*Proof.* Let $\boldsymbol{\alpha}^* \in \arg\max_{\boldsymbol{\alpha} \in \mathcal{A}} f(\boldsymbol{\theta}, \boldsymbol{\alpha})$. By Lemma A.3, for any scalar $\tau$ and direction $d$, there exists $\boldsymbol{\alpha}^*(\tau) \in \arg\max_{\boldsymbol{\alpha}} f(\boldsymbol{\theta} + \tau d, \boldsymbol{\alpha})$ such that

$$\|\boldsymbol{\alpha}^*(\tau) - \boldsymbol{\alpha}^*\| \leq \frac{L_{12}}{2\mu}\tau\|d\|.$$

To find the directional derivative of $g(\cdot)$, we compute

$$\begin{aligned}
g(\boldsymbol{\theta} + \tau d) - g(\boldsymbol{\theta}) &= f(\boldsymbol{\theta} + \tau d, \boldsymbol{\alpha}^*(\tau)) - f(\boldsymbol{\theta}, \boldsymbol{\alpha}^*) \\
&= \tau \nabla_{\boldsymbol{\theta}} f(\boldsymbol{\theta}, \boldsymbol{\alpha}^*)^T d + \underbrace{\nabla_{\boldsymbol{\alpha}} f(\boldsymbol{\theta}, \boldsymbol{\alpha}^*)^T}_{0}(\boldsymbol{\alpha}^*(\tau) - \boldsymbol{\alpha}^*) + \mathcal{O}(\tau^2),
\end{aligned}$$

where the second equality holds by writing the Taylor series expansion of $f(\cdot)$. Thus, by definition of the directional derivative of $g(\cdot)$, we obtain

$$g'(\boldsymbol{\theta}; d) = \lim_{\tau \to 0^+} \frac{g(\boldsymbol{\theta} + \tau d) - g(\boldsymbol{\theta})}{\tau} = \nabla_{\boldsymbol{\theta}} f(\boldsymbol{\theta}, \boldsymbol{\alpha}^*)^T d. \tag{19}$$

Note that this relationship holds for any $d$. Thus, $\nabla g(\boldsymbol{\theta}) = \nabla_{\boldsymbol{\theta}} f(\boldsymbol{\theta}, \boldsymbol{\alpha}^*)$ for any $\boldsymbol{\alpha}^* \in \arg\max_{\boldsymbol{\alpha} \in \mathcal{A}} f(\boldsymbol{\theta}, \boldsymbol{\alpha}) = A(\boldsymbol{\theta})$. Interestingly, the directional derivative does not depend on the choice of $\boldsymbol{\alpha}^*$. This means that $\nabla_{\boldsymbol{\theta}} f(\boldsymbol{\theta}, \boldsymbol{\alpha}_1) = \nabla_{\boldsymbol{\theta}} f(\boldsymbol{\theta}, \boldsymbol{\alpha}_2)$ for any $\boldsymbol{\alpha}_1$ and $\boldsymbol{\alpha}_2$ in $\arg\max_{\boldsymbol{\alpha} \in \mathcal{A}} f(\boldsymbol{\theta}, \boldsymbol{\alpha})$.

We finally show that function $g$ is Lipschitz smooth. Let $\boldsymbol{\alpha}_1^* \in A(\boldsymbol{\theta}_1)$ and $\boldsymbol{\alpha}_2^* = \arg\min_{\boldsymbol{\alpha} \in A(\boldsymbol{\theta}_2)} \|\boldsymbol{\alpha} - \boldsymbol{\alpha}_1^*\|^2 \in A(\boldsymbol{\theta}_2)$, then

$$\begin{aligned}
\|\nabla g(\boldsymbol{\theta}_1) - \nabla g(\boldsymbol{\theta}_2)\| &= \|\nabla_{\boldsymbol{\theta}} f(\boldsymbol{\theta}_1, \boldsymbol{\alpha}_1^*) - \nabla_{\boldsymbol{\theta}} f(\boldsymbol{\theta}_2, \boldsymbol{\alpha}_2^*)\| \\
&= \|\nabla_{\boldsymbol{\theta}} f(\boldsymbol{\theta}_1, \boldsymbol{\alpha}_1^*) - \nabla_{\boldsymbol{\theta}} f(\boldsymbol{\theta}_2, \boldsymbol{\alpha}_1^*) + \nabla_{\boldsymbol{\theta}} f(\boldsymbol{\theta}_2, \boldsymbol{\alpha}_1^*) - \nabla_{\boldsymbol{\theta}} f(\boldsymbol{\theta}_2, \boldsymbol{\alpha}_2^*)\| \\
&\leq L_{11}\|\boldsymbol{\theta}_1 - \boldsymbol{\theta}_2\| + L_{12}\|\boldsymbol{\alpha}_1^* - \boldsymbol{\alpha}_2^*\| \\
&\leq \left(L_{11} + \frac{L_{12}^2}{2\mu}\right)\|\boldsymbol{\theta}_1 - \boldsymbol{\theta}_2\|,
\end{aligned}$$

where the last inequality holds by Lemma A.3.

$\square$

## A.2 Proof of Theorem 3.4

Using Lemma A.5 and Assumption 3.3, we can define

$$g_{\boldsymbol{\theta}} \triangleq \max_{\boldsymbol{\theta} \in \Theta} \|\nabla g(\boldsymbol{\theta})\| \quad \text{and} \quad g_{max} \triangleq \max\{g_{\boldsymbol{\theta}}, 1\}. \tag{20}$$

The next result shows that the inner loop in Algorithm 1 computes an approximate gradient of $g(\cdot)$. In other words, $\nabla_{\boldsymbol{\theta}} f(\boldsymbol{\theta}_t, \boldsymbol{\alpha}_{t+1}) \approx \nabla g(\boldsymbol{\theta}_t)$.

**Lemma A.6.** *Define* $\kappa = \frac{L_{22}}{\mu} \geq 1$ *and* $\rho = 1 - \frac{1}{\kappa} < 1$ *and assume* $g(\boldsymbol{\theta}_t) - f(\boldsymbol{\theta}_t, \boldsymbol{\alpha}_0(\boldsymbol{\theta}_t)) < \Delta$, *then for any prescribed* $\varepsilon \in (0, 1)$ *if we choose $K$ large enough such that*

$$K \geq N_K(\varepsilon) \triangleq \frac{1}{\log 1/\rho}\left(4\log(1/\varepsilon) + \log(2^{15}\bar{L}^6\bar{R}^6\Delta/L^2\mu)\right), \tag{21}$$

*where* $\bar{L} = \max\{L_{12}, L_{22}, L, g_{max}, 1\}$ *and* $\bar{R} = \max\{R, 1\}$, *then the error* $e_t \triangleq \nabla_{\boldsymbol{\theta}} f(\boldsymbol{\theta}_t, \boldsymbol{\alpha}_K(\boldsymbol{\theta}_t)) - \nabla g(\boldsymbol{\theta}_t)$ *has a norm*

$$\|e_t\| \leq \delta \triangleq \frac{L\varepsilon^2}{2^6 R(g_{max} + LR)^2} \quad \text{and} \quad \|\nabla_{\boldsymbol{\alpha}} f(\boldsymbol{\theta}_t, \boldsymbol{\alpha}_K(\boldsymbol{\theta}_t))\| \leq \varepsilon. \tag{22}$$

*Proof.* First of all, Lemma A.4 implies that

$$g(\boldsymbol{\theta}_t) - f(\boldsymbol{\theta}_t, \boldsymbol{\alpha}_K(\boldsymbol{\theta}_t)) \leq \rho^K \Delta. \tag{23}$$

Thus, using the QG result of Lemma A.2, we know that there exists an $\boldsymbol{\alpha}^* \in A(\boldsymbol{\theta}_t)$ such that

$$\|\boldsymbol{\alpha}_K(\boldsymbol{\theta}_t) - \boldsymbol{\alpha}^*\| \leq \rho^{K/2}\sqrt{\frac{\Delta}{2\mu}} \tag{24}$$

Thus,

$$\|e_t\| = \|\nabla_{\boldsymbol{\theta}} f(\boldsymbol{\theta}_t, \boldsymbol{\alpha}_K(\boldsymbol{\theta}_t)) - \nabla g(\boldsymbol{\theta})\| \leq L_{12}\|\boldsymbol{\alpha}_K(\boldsymbol{\theta}_t) - \boldsymbol{\alpha}^*\|$$

$$\leq L_{12}\rho^{K/2}\sqrt{\frac{\Delta}{2\mu}}$$

$$\leq \frac{L\varepsilon^2}{2^6 R(g_{max} + LR)^2}, \tag{25}$$

where the last inequality holds by our choice of $K$ which yields

$$\log\left(1/\rho\right)^K \geq \log\left(1/\varepsilon\right)^4 + \log\left(2^{15}\bar{L}^6\bar{R}^6\Delta/L^2\mu\right) = \log\left(2^{15}\bar{L}^6\bar{R}^6\Delta/L^2\mu\varepsilon^4\right)$$

which implies,

$$\rho^K \leq \frac{2L^2\varepsilon^4\mu}{2^{12}\Delta\bar{R}^2\bar{L}^2(2\bar{L}\bar{R})^4} \leq \frac{2L^2\varepsilon^4\mu}{2^{12}\Delta R^2\bar{L}^2(\bar{L}+\bar{L}R)^4} \leq \frac{2L^2\varepsilon^4\mu}{2^{12}\Delta R^2\bar{L}^2(g_{max}+LR)^4}.$$

Here the second inequality holds since $\bar{R} \geq 1$, and the third inequality holds since $g_{max} \leq \bar{L}$.

To prove the argument of the Lemma, note that

$$\|\nabla_{\boldsymbol{\alpha}} f(\boldsymbol{\theta}_t, \boldsymbol{\alpha}_K(\boldsymbol{\theta}_t)) - \underbrace{\nabla_{\boldsymbol{\alpha}} f(\boldsymbol{\theta}_t, \boldsymbol{\alpha}^*)}_{0}\| \leq L_{22}\|\boldsymbol{\alpha}_K(\boldsymbol{\theta}_t) - \boldsymbol{\alpha}^*\| \leq L_{22}\rho^{K/2}\sqrt{\frac{\Delta}{2\mu}} \leq \varepsilon, \tag{26}$$

where the last inequality holds by our choice of $K$ which yields

$$\rho^K \leq \left(\frac{\varepsilon^2\mu}{\bar{L}^2\Delta}\right)\underbrace{\left(\frac{\varepsilon^2L^2}{2^{15}\bar{L}^4\bar{R}^4}\right)}_{\leq 1} \leq \frac{\varepsilon^2\mu}{\bar{L}^2\Delta}.$$

Here the second inequality holds since $\varepsilon < 1$, $\bar{L}, \bar{R} \geq 1$, and $L \leq \bar{L}$.

$\square$

The above lemma implies that Algorithm 1 behaves similar to the simple vanilla gradient descent method applied to problem (8).

Notice that the assumption $g(\boldsymbol{\theta}_t) - f(\boldsymbol{\theta}_t, \boldsymbol{\alpha}_0(\boldsymbol{\theta}_t)) \leq \Delta$, $\forall t$ could be justified by Lemma A.3. More specifically, by Lemma A.3,

$$\|\boldsymbol{\alpha}_{t+1} - \boldsymbol{\alpha}_t\| \leq \frac{L_{12}}{2\mu}\|\boldsymbol{\theta}_{t+1} - \boldsymbol{\theta}_t\|,$$

where $\boldsymbol{\alpha}_{t+1} \triangleq \arg\max_{\boldsymbol{\alpha}} f(\boldsymbol{\theta}_{t+1}, \boldsymbol{\alpha})$ and $\boldsymbol{\alpha}_t \triangleq \arg\max_{\boldsymbol{\alpha}} f(\boldsymbol{\theta}_t, \boldsymbol{\alpha})$. Hence, the difference between consecutive optimal solutions computed by the inner loop of the algorithm, are upper bounded by the difference between corresponding $\boldsymbol{\theta}$'s. Since $\Theta$ is a compact set, we can find an upper bound $\Delta$ such that $g(\boldsymbol{\theta}_t) - f(\boldsymbol{\theta}_t, \alpha_0(\boldsymbol{\theta}_t)) \leq \Delta$, for all $t$. We are now ready to show Theorem 3.4

*Proof.* We start by defining
$$\Delta_g = g(\boldsymbol{\theta}_0) - g^*,$$
where $g^* \triangleq \min_{\boldsymbol{\theta}} g(\boldsymbol{\theta})$ is the optimal value of $g$. Note that by the compactness assumption of the set $\Theta$, we have $\Delta_g = g(\theta_0) - g^* < \infty$.

Based on the projection property, we know that

$$\left\langle \boldsymbol{\theta}_t - \frac{1}{L}\nabla_{\boldsymbol{\theta}} f(\boldsymbol{\theta}_t, \boldsymbol{\alpha}_{t+1}) - \boldsymbol{\theta}_{t+1}, \boldsymbol{\theta} - \boldsymbol{\theta}_{t+1} \right\rangle \leq 0 \quad \forall \ \boldsymbol{\theta} \in \Theta.$$

Therefore, by setting $\boldsymbol{\theta} = \boldsymbol{\theta}_t$, we get

$$\left\langle \nabla_{\boldsymbol{\theta}} f(\boldsymbol{\theta}_t, \boldsymbol{\alpha}_{t+1}), \boldsymbol{\theta}_{t+1} - \boldsymbol{\theta}_t \right\rangle \leq -L\|\boldsymbol{\theta}_t - \boldsymbol{\theta}_{t+1}\|^2,$$

which implies

$$\langle \nabla_{\boldsymbol{\theta}} f(\boldsymbol{\theta}_t, \boldsymbol{\alpha}^*(\boldsymbol{\theta}_t)), \boldsymbol{\theta}_{t+1} - \boldsymbol{\theta}_t \rangle \leq -L\|\boldsymbol{\theta}_t - \boldsymbol{\theta}_{t+1}\|^2 + \langle \nabla_{\boldsymbol{\theta}} f(\boldsymbol{\theta}_t, \boldsymbol{\alpha}^*(\boldsymbol{\theta}_t)) - \nabla_{\boldsymbol{\theta}} f(\boldsymbol{\theta}_t, \boldsymbol{\alpha}_{t+1}), \boldsymbol{\theta}_{t+1} - \boldsymbol{\theta}_t \rangle$$
$$= -L\|\boldsymbol{\theta}_t - \boldsymbol{\theta}_{t+1}\|^2 + \langle e_t, \boldsymbol{\theta}_t - \boldsymbol{\theta}_{t+1} \rangle$$

(27)

where $\boldsymbol{\alpha}^*(\boldsymbol{\theta}_t) \in \arg\max_{\boldsymbol{\alpha} \in \mathcal{A}} f(\boldsymbol{\theta}_t, \boldsymbol{\alpha})$ and $e_t \triangleq \nabla_{\boldsymbol{\theta}} f(\boldsymbol{\theta}_t, \boldsymbol{\alpha}_{t+1}) - \nabla_{\boldsymbol{\theta}} f(\boldsymbol{\theta}_t, \boldsymbol{\alpha}^*(\boldsymbol{\theta}_t))$. By Taylor expansion, we have

$$g(\boldsymbol{\theta}_{t+1}) \leq g(\boldsymbol{\theta}_t) + \langle \nabla_{\boldsymbol{\theta}} f(\boldsymbol{\theta}_t, \boldsymbol{\alpha}^*(\boldsymbol{\theta}_t)), \boldsymbol{\theta}_{t+1} - \boldsymbol{\theta}_t \rangle + \frac{L}{2}\|\boldsymbol{\theta}_{t+1} - \boldsymbol{\theta}_t\|^2$$
$$\leq g(\boldsymbol{\theta}_t) - \frac{L}{2}\|\boldsymbol{\theta}_{t+1} - \boldsymbol{\theta}_t\|^2 + \langle e_t, \boldsymbol{\theta}_t - \boldsymbol{\theta}_{t+1} \rangle.$$

(28)

where the last inequality holds by (27). Moreover, by the projection property, we know that

$$\langle \nabla_{\boldsymbol{\theta}} f(\boldsymbol{\theta}_t, \boldsymbol{\alpha}_{t+1}), \boldsymbol{\theta} - \boldsymbol{\theta}_{t+1} \rangle \geq L\langle \boldsymbol{\theta}_t - \boldsymbol{\theta}_{t+1}, \boldsymbol{\theta} - \boldsymbol{\theta}_{t+1} \rangle \quad \forall \boldsymbol{\theta} \in \Theta,$$

which implies

$$\langle \nabla_{\boldsymbol{\theta}} f(\boldsymbol{\theta}_t, \boldsymbol{\alpha}_{t+1}), \boldsymbol{\theta} - \boldsymbol{\theta}_t \rangle \geq \langle \nabla_{\boldsymbol{\theta}} f(\boldsymbol{\theta}_t, \boldsymbol{\alpha}_{t+1}), \boldsymbol{\theta}_{t+1} - \boldsymbol{\theta}_t \rangle + L\langle \boldsymbol{\theta}_t - \boldsymbol{\theta}_{t+1}, \boldsymbol{\theta} - \boldsymbol{\theta}_{t+1} \rangle$$

$$\geq -(g_{max} + 2LR + \|e_t\|)\|\boldsymbol{\theta}_{t+1} - \boldsymbol{\theta}_t\|$$

(29)

$$\geq -2(g_{max} + LR)\|\boldsymbol{\theta}_{t+1} - \boldsymbol{\theta}_t\|.$$

Here the second inequality holds by Cauchy-Schwartz, the definition of $e_t$ and our assumption that $\Theta \subseteq \mathcal{B}_R$. Moreover, the last inequality holds by our choice of $K$ in Lemma A.6 which yields

$$\|e_t\| = \|\nabla_{\boldsymbol{\theta}} f(\boldsymbol{\theta}_t, \boldsymbol{\alpha}_K(\boldsymbol{\theta}_t)) - \nabla g(\boldsymbol{\theta})\| \tag{30}$$
$$\leq L_{12}\|\boldsymbol{\alpha}_K(\boldsymbol{\theta}_t) - \boldsymbol{\alpha}^*\|$$
$$\leq L_{12}\rho^{K/2}\sqrt{\frac{\Delta}{2\mu}}$$
$$\leq 1 \tag{31}$$
$$\leq g_{max}. \tag{32}$$

Hence,

$$-\mathcal{X}_t \geq -2(g_{max} + LR)\|\boldsymbol{\theta}_{t+1} - \boldsymbol{\theta}_t\|,$$

or equivalently

$$\|\boldsymbol{\theta}_{t+1} - \boldsymbol{\theta}_t\| \geq \frac{\mathcal{X}_t}{2(g_{max} + LR)}. \tag{33}$$

Combined with (28), we get

$$g(\boldsymbol{\theta}_{t+1}) - g(\boldsymbol{\theta}_t) \leq -\frac{L}{8}\frac{\mathcal{X}_t^2}{(g_{max} + LR)^2} + 2\|e_t\|R,$$

where the inequality holds by using Cauchy Schwartz and our assumption that $\Theta$ is in a ball of radius $R$. Hence,

$$\frac{1}{T}\sum_{t=0}^{T-1} \mathcal{X}_t^2 \leq \frac{8\Delta_g(g_{max} + LR)^2}{LT} + \frac{16\delta R(g_{max} + LR)^2}{L}$$

$$\leq \frac{\varepsilon^2}{2},$$

where the last inequality holds by using Lemma A.6 and choosing $K$ and $T$:

$$T \geq N_T \triangleq \frac{32\Delta_g(g_{max} + LR)^2}{L\varepsilon^2}, \quad K \geq N_K(\varepsilon) \triangleq \frac{1}{\log 1/\rho}\left(4\log(1/\varepsilon) + \log(2^{15}\bar{L}^6\bar{R}^6\Delta/L^2\mu)\right).$$

Therefore, using Lemma A.6, there exists at least one index $\widehat{t}$ for which

$$\mathcal{X}_{\widehat{t}} \le \varepsilon \quad \text{and} \quad \|\nabla_{\boldsymbol{\alpha}} f(\boldsymbol{\theta}_{\widehat{t}}, \boldsymbol{\alpha}_{\widehat{t}+1})\| \le \varepsilon. \tag{34}$$

This completes the proof of the theorem.

$\square$

# B Algorithmic details and proofs for the results in Section 4

## B.1 Accelerated Projected Gradient Ascent Subroutine Used in Algorithm 2

---
**Algorithm 3** APGA: Accelerated Projected Gradient Ascent with Restart

**Require:** Constants $\boldsymbol{\alpha}_t$, $\boldsymbol{\theta}_t$, $\eta$, $K$, and $N$.

---
1: **for** $k = 0, \ldots, \lfloor K/N \rfloor$ **do**
2:     Set $\gamma_1 = 1$
3:     **if** $k = 0$ **then** $\mathbf{y}_1 = \boldsymbol{\alpha}_t$ **else** $\mathbf{y}_1 = \mathbf{x}_N$
4:     **for** $i = 1, 2, \ldots, N$ **do**
5:         Set $\mathbf{x}_i = \text{proj}_{\mathcal{A}}\big(\mathbf{y}_i + \eta\nabla_{\mathbf{y}} f_{\lambda}(\boldsymbol{\theta}_t, \mathbf{y}_i)\big)$
6:         Set $\gamma_{i+1} = \dfrac{1 + \sqrt{1 + 4\gamma_i^2}}{2}$
7:         $\mathbf{y}_{i+1} = \mathbf{x}_i + \left(\dfrac{\gamma_i - 1}{\gamma_{i+1}}\right)(\mathbf{x}_i - \mathbf{x}_{i-1})$
8:     **end for**
9: **end for**
10: Return $\mathbf{x}_N$

---

## B.2 Frank–Wolfe update rule for Step 3 in Algorithm 2

In Step 3 of Algorithm 2, instead of projected gradient descent discussed in the main body, we can also run one step of Frank–Wolfe method. More precisely, we can set

$$\boldsymbol{\theta}_{t+1} = \boldsymbol{\theta}_t + \frac{\mathcal{X}_t}{\widetilde{L}}\widehat{s}_t,$$

where

$$\mathcal{X}_t \triangleq -\min_s \; \langle \nabla_{\boldsymbol{\theta}} f_{\lambda}(\boldsymbol{\theta}_t, \boldsymbol{\alpha}_{t+1}), s \rangle$$
$$\text{s.t. } \boldsymbol{\theta}_t + s \in \Theta, \|s\| \le 1, \tag{35}$$

and

$$\widehat{s}_t \triangleq \arg\min_s \; \langle \nabla_{\boldsymbol{\theta}} f_{\lambda}(\boldsymbol{\theta}_t, \boldsymbol{\alpha}_K(\boldsymbol{\theta}_t)), s \rangle$$
$$\text{s.t. } \boldsymbol{\theta}_t + s \in \Theta, \|s\| \le 1. \tag{36}$$

is the first order descent direction. In the unconstrained case, the descent direction is $\widehat{s}_t = -\nabla_{\boldsymbol{\theta}} f_{\lambda}(\boldsymbol{\theta}_t, \boldsymbol{\alpha}_{t+1})$, which becomes the same as the gradient descent step.

## B.3 Smoothness of function $g_{\lambda}(\cdot)$

**Lemma B.1.** *Under Assumption 2.5 and Assumption 4.1, the function $g_{\lambda}$ is L-Lipschitz smooth with $L = L_{11} + \dfrac{L_{12}^2}{\lambda}$.*

*Proof.* First notice that the differentiability of the function $g_{\lambda}(\cdot)$ follows directly from Danskin's Theorem [4]. It remains to show that $g_{\lambda}$ is a Lipschitz smooth function. Let

$$\boldsymbol{\alpha}_1^* \triangleq \arg\max_{\boldsymbol{\alpha} \in \mathcal{A}} \; f_{\lambda}(\boldsymbol{\theta}_1, \boldsymbol{\alpha}) \quad \text{and} \quad \boldsymbol{\alpha}_2^* \triangleq \arg\max_{\boldsymbol{\alpha} \in \mathcal{A}} \; f_{\lambda}(\boldsymbol{\theta}_2, \boldsymbol{\alpha}).$$

Then by strong convexity of $-f_\lambda(\boldsymbol{\theta}, \cdot)$, we have

$$f_\lambda(\boldsymbol{\theta}_2, \boldsymbol{\alpha}_2^*) \leq f_\lambda(\boldsymbol{\theta}_2, \boldsymbol{\alpha}_1^*) + \langle \nabla_{\boldsymbol{\alpha}} f_\lambda(\boldsymbol{\theta}_2, \boldsymbol{\alpha}_1^*), \boldsymbol{\alpha}_2^* - \boldsymbol{\alpha}_1^* \rangle - \frac{\lambda}{2} \|\boldsymbol{\alpha}_2^* - \boldsymbol{\alpha}_1^*\|^2,$$

and

$$f_\lambda(\boldsymbol{\theta}_2, \boldsymbol{\alpha}_1^*) \leq f_\lambda(\boldsymbol{\theta}_2, \boldsymbol{\alpha}_2^*) + \underbrace{\langle \nabla_{\boldsymbol{\alpha}} f_\lambda(\boldsymbol{\theta}_2, \boldsymbol{\alpha}_2^*), \boldsymbol{\alpha}_1^* - \boldsymbol{\alpha}_2^* \rangle}_{\leq 0, \text{ by optimality of } \boldsymbol{\alpha}_2^*} - \frac{\lambda}{2} \|\boldsymbol{\alpha}_2^* - \boldsymbol{\alpha}_1^*\|^2.$$

Adding the two inequalities, we get

$$\langle \nabla_{\boldsymbol{\alpha}} f_\lambda(\boldsymbol{\theta}_2, \boldsymbol{\alpha}_1^*), \boldsymbol{\alpha}_2^* - \boldsymbol{\alpha}_1^* \rangle \geq \lambda \|\boldsymbol{\alpha}_2^* - \boldsymbol{\alpha}_1^*\|^2. \tag{37}$$

Moreover, due to optimality of $\boldsymbol{\alpha}_1^*$, we have

$$\langle \nabla_{\boldsymbol{\alpha}} f_\lambda(\boldsymbol{\theta}_1, \boldsymbol{\alpha}_1^*), \boldsymbol{\alpha}_2^* - \boldsymbol{\alpha}_1^* \rangle \leq 0. \tag{38}$$

Combining (37) and (38) we obtain

$$\begin{aligned} \lambda \|\boldsymbol{\alpha}_2^* - \boldsymbol{\alpha}_1^*\|^2 &\leq \langle \nabla_{\boldsymbol{\alpha}} f_\lambda(\boldsymbol{\theta}_2, \boldsymbol{\alpha}_1^*) - \nabla_{\boldsymbol{\alpha}} f_\lambda(\boldsymbol{\theta}_1, \boldsymbol{\alpha}_1^*), \boldsymbol{\alpha}_2^* - \boldsymbol{\alpha}_1^* \rangle \\ &\leq L_{12} \|\boldsymbol{\theta}_1 - \boldsymbol{\theta}_2\| \|\boldsymbol{\alpha}_2^* - \boldsymbol{\alpha}_1^*\|, \end{aligned} \tag{39}$$

where the last inequality holds by Cauchy-Schwartz and the Lipschtizness assumption. We finally show that $g_\lambda$ is Lipschitz smooth.

$$\begin{aligned} \|\nabla g_\lambda(\boldsymbol{\theta}_1) - \nabla g_\lambda(\boldsymbol{\theta}_2)\| &= \|\nabla_{\boldsymbol{\theta}} f_\lambda(\boldsymbol{\theta}_1, \boldsymbol{\alpha}_1^*) - \nabla_{\boldsymbol{\theta}} f_\lambda(\boldsymbol{\theta}_2, \boldsymbol{\alpha}_2^*)\| \\ &= \|\nabla_{\boldsymbol{\theta}} f_\lambda(\boldsymbol{\theta}_1, \boldsymbol{\alpha}_1^*) - \nabla_{\boldsymbol{\theta}} f_\lambda(\boldsymbol{\theta}_2, \boldsymbol{\alpha}_1^*) + \nabla_{\boldsymbol{\theta}} f_\lambda(\boldsymbol{\theta}_2, \boldsymbol{\alpha}_1^*) - \nabla_{\boldsymbol{\theta}} f_\lambda(\boldsymbol{\theta}_2, \boldsymbol{\alpha}_2^*)\| \\ &\leq L_{11} \|\boldsymbol{\theta}_1 - \boldsymbol{\theta}_2\| + L_{12} \|\boldsymbol{\alpha}_1^* - \boldsymbol{\alpha}_2^*\| \\ &\leq \left( L_{11} + \frac{L_{12}^2}{\lambda} \right) \|\boldsymbol{\theta}_1 - \boldsymbol{\theta}_2\|, \end{aligned}$$

where the last inequality holds by (39). $\qquad \square$

Algorithm 2 solves the inner maximization problem using accelerated projected gradient descent (outlined in Algorithm 3). The next lemma is known for accelerated projected gradient descent when applied to strongly convex functions.

**Lemma B.2.** *Assume $h(\mathbf{x})$ is $\lambda$-strongly convex and $L$-smooth. Then, applying accelerated projected gradient descent algorithm [3] with step-size $1/L$ and restart parameter $N \triangleq \sqrt{8L/\lambda} - 1$ for $K$ iterations, we get $\mathbf{x}_K$ such that*

$$h(\mathbf{x}_K) - h(\mathbf{x}^*) \leq \left( \frac{1}{2} \right)^{K/N} (h(\mathbf{x}_0) - h(\mathbf{x}^*)), \tag{40}$$

*where $\mathbf{x}^* \triangleq \arg\min_{\mathbf{x} \in \mathcal{F}} h(\mathbf{x})$.*

*Proof.* According to [3, Theorem 4.4], we have

$$\begin{aligned} h(\mathbf{x}_{iN}) - h(\mathbf{x}^*) &\leq \frac{2L}{(N+1)^2} \|\mathbf{x}_{(i-1)N} - \mathbf{x}^*\|^2 \\ &\leq \frac{4L}{\lambda(N+1)^2} \big( h(\mathbf{x}_{(i-1)N}) - h(\mathbf{x}^*) \big) \\ &\leq \frac{1}{2} \big( h(\mathbf{x}_{(i-1)N}) - h(\mathbf{x}^*) \big), \end{aligned} \tag{41}$$

where the second inequality holds by strong convexity of $h$ and the optimality condition of $x^*$, and the last inequality holds by our choice of $N$. This yields,

$$h(\mathbf{x}_K) - h(\mathbf{x}^*) \leq (\frac{1}{2})^{K/N} \big( h(\mathbf{x}_0) - h(\mathbf{x}^*) \big), \tag{42}$$

which completes our proof. $\qquad \square$

## B.4 Proof of Theorem 4.2

We first show that the inner loop in Algorithm 2 computes an approximate gradient of $g_\lambda(\cdot)$. In other words, $\nabla_{\boldsymbol{\theta}} f_\lambda(\boldsymbol{\theta}_t, \boldsymbol{\alpha}_{t+1}) \approx \nabla g_\lambda(\boldsymbol{\theta}_t)$.

**Lemma B.3.** *Define* $\kappa = \dfrac{L_{22}}{\lambda} \geq 1$ *and assume* $g_\lambda(\boldsymbol{\theta}_t) - f_\lambda(\boldsymbol{\theta}_t, \boldsymbol{\alpha}_0(\boldsymbol{\theta}_t)) < \Delta$, *then for any prescribed* $\varepsilon \in (0, 1)$ *if we choose* $K$ *large enough such that*

$$K \geq N_K(\varepsilon) \triangleq \frac{\sqrt{8\kappa}}{\log 2} \left( 4 \log(1/\varepsilon) + \log(2^{17} \bar{L}^6 \bar{R}^6 \Delta / L^2 \lambda) \right), \tag{43}$$

*where* $\bar{L} \triangleq \max\{L_{12}, L_{22}, L, g_{max}, 1\}$ *and* $\bar{R} = \max\{R, 1\}$, *then the error* $e_t \triangleq \nabla_{\boldsymbol{\theta}} f_\lambda(\boldsymbol{\theta}_t, \boldsymbol{\alpha}_K(\boldsymbol{\theta}_t)) - \nabla g_\lambda(\boldsymbol{\theta})$ *has a norm*

$$\|e_t\| \leq \delta \triangleq \frac{L\varepsilon^2}{2^6 R(g_{max} + LR)^2} \tag{44}$$

*and*

$$\frac{\varepsilon}{2} \geq \mathcal{Y}_{t,K} \triangleq \max_{s} \; \langle \nabla_{\boldsymbol{\alpha}} f_\lambda(\boldsymbol{\theta}_t, \boldsymbol{\alpha}_K(\boldsymbol{\theta}_t)), s \rangle \\ \text{s.t.} \quad \boldsymbol{\alpha}_K(\boldsymbol{\theta}_t) + s \in \mathcal{A}, \|s\| \leq 1 \tag{45}$$

*Proof.* Starting from Lemma B.2, we have that

$$g_\lambda(\boldsymbol{\theta}_t) - f_\lambda(\boldsymbol{\theta}_t, \boldsymbol{\alpha}_K(\boldsymbol{\theta}_t)) \leq \frac{1}{2^{\frac{K}{\sqrt{8\kappa}}}} \Delta. \tag{46}$$

Let $\boldsymbol{\alpha}^*(\boldsymbol{\theta}_t) \triangleq \arg\max_{\boldsymbol{\alpha} \in \mathcal{A}} \; f_\lambda(\boldsymbol{\theta}_t, \boldsymbol{\alpha})$. Then by strong convexity of $-f(\boldsymbol{\theta}_t, \cdot)$, we get

$$\frac{\lambda}{2} \|\boldsymbol{\alpha}_K(\boldsymbol{\theta}_t) - \boldsymbol{\alpha}^*(\boldsymbol{\theta}_t)\|^2 \leq g_\lambda(\boldsymbol{\theta}_t) - f_\lambda(\boldsymbol{\theta}_t, \boldsymbol{\alpha}_K(\boldsymbol{\theta}_t)) \leq \frac{1}{2^{\frac{K}{\sqrt{8\kappa}}}} \Delta. \tag{47}$$

Combined with the Lipschitz smoothness property of the objective, we obtain

$$\begin{aligned} \|e_t\| &= \|\nabla_{\boldsymbol{\theta}} f_\lambda(\boldsymbol{\theta}_t, \boldsymbol{\alpha}_K(\boldsymbol{\theta}_t)) - \nabla g_\lambda(\boldsymbol{\theta}_t)\| \\ &= \|\nabla_{\boldsymbol{\theta}} f_\lambda(\boldsymbol{\theta}_t, \boldsymbol{\alpha}_K(\boldsymbol{\theta}_t)) - \nabla_{\boldsymbol{\theta}} f_\lambda(\boldsymbol{\theta}_t, \boldsymbol{\alpha}^*(\boldsymbol{\theta}_t))\| \\ &\leq L_{12} \|\boldsymbol{\alpha}_K(\boldsymbol{\theta}_t) - \boldsymbol{\alpha}^*(\boldsymbol{\theta}_t)\| \\ &\leq \frac{L_{12}}{2^{K/2\sqrt{8\kappa}}} \sqrt{\frac{2\Delta}{\lambda}} \\ &\leq \frac{L\varepsilon^2}{2^6 R(g_{max} + LR)^2} \end{aligned} \tag{48}$$

where the second inequality uses (47), and the third inequality uses the choice of $K$ in (43) which yields yields

$$\log\left(2^{K/\sqrt{8\kappa}}\right) \geq \log\left(1/\varepsilon\right)^4 + \log\left(2^{17} \bar{L}^6 \bar{R}^6 \Delta / L^2 \lambda\right) = \log\left(2^{17} \bar{L}^6 \bar{R}^6 \Delta / L^2 \lambda \varepsilon^4\right)$$

which implies,

$$\left(\frac{1}{2}\right)^{K/2\sqrt{8\kappa}} \leq \frac{L\varepsilon^2 \sqrt{\lambda}}{2^6 \sqrt{2\Delta} \bar{R}\bar{L}(2\bar{L}\bar{R})^2} \leq \frac{L\varepsilon^2 \sqrt{\lambda}}{2^6 \sqrt{2\Delta} R\bar{L}(\bar{L} + \bar{L}R)^2} \leq \frac{L\varepsilon^2 \sqrt{\lambda}}{2^6 \sqrt{2\Delta} R\bar{L}(g_{max} + LR)^2}.$$

Here the second inequality holds since $\bar{R} \geq 1$, and the third inequality holds since $g_{max} \leq \bar{L}$. To prove the second argument of the lemma, we also use the Lipschitz smoothness property of the

objective to get

$$\langle \nabla_{\boldsymbol{\alpha}} f_\lambda(\boldsymbol{\theta}_t, \boldsymbol{\alpha}_K(\boldsymbol{\theta}_t)), s \rangle = \langle \nabla_{\boldsymbol{\alpha}} f_\lambda(\boldsymbol{\theta}_t, \boldsymbol{\alpha}_K(\boldsymbol{\theta}_t)) - \nabla_{\boldsymbol{\alpha}} f_\lambda(\boldsymbol{\theta}_t, \boldsymbol{\alpha}^*(\boldsymbol{\theta}_t)), s \rangle + \langle \nabla_{\boldsymbol{\alpha}} f_\lambda(\boldsymbol{\theta}_t, \boldsymbol{\alpha}^*(\boldsymbol{\theta}_t)), s \rangle$$
$$\leq \|\nabla_{\boldsymbol{\alpha}} f_\lambda(\boldsymbol{\theta}_t, \boldsymbol{\alpha}_K(\boldsymbol{\theta}_t)) - \nabla_{\boldsymbol{\alpha}} f_\lambda(\boldsymbol{\theta}_t, \boldsymbol{\alpha}^*(\boldsymbol{\theta}_t))\| \|s\| + \langle \nabla_{\boldsymbol{\alpha}} f_\lambda(\boldsymbol{\theta}_t, \boldsymbol{\alpha}^*(\boldsymbol{\theta}_t)), s \rangle$$
$$\leq (L_{22} + \lambda) \|\boldsymbol{\alpha}^*(\boldsymbol{\theta}_t) - \boldsymbol{\alpha}_K(\boldsymbol{\theta}_t)) \| \|s\| + \langle \nabla_{\boldsymbol{\alpha}} f_\lambda(\boldsymbol{\theta}_t, \boldsymbol{\alpha}^*(\boldsymbol{\theta}_t)), s \rangle.$$
$$\leq 2L_{22} \|\boldsymbol{\alpha}^*(\boldsymbol{\theta}_t) - \boldsymbol{\alpha}_K(\boldsymbol{\theta}_t)) \| \|s\| + \langle \nabla_{\boldsymbol{\alpha}} f_\lambda(\boldsymbol{\theta}_t, \boldsymbol{\alpha}^*(\boldsymbol{\theta}_t)), s \rangle,$$

(49)

where the second inequality holds by our Lipschitzness assumption and the last inequality holds by our assumption that $L_{22}/\lambda \geq 1$. Moreover,

$$\begin{array}{ll} \min_s & -\langle \nabla_{\boldsymbol{\alpha}} f_\lambda(\boldsymbol{\theta}_t, \boldsymbol{\alpha}^*(\boldsymbol{\theta}_t)), s \rangle \\ \text{s.t.} & \boldsymbol{\alpha}_K(\boldsymbol{\theta}_t) + s \in \mathcal{A}, \|s\| \leq 1 \end{array} = \begin{array}{ll} \min_{\boldsymbol{\alpha}} & -\langle \nabla_{\boldsymbol{\alpha}} f_\lambda(\boldsymbol{\theta}_t, \boldsymbol{\alpha}^*(\boldsymbol{\theta}_t)), \boldsymbol{\alpha} - \boldsymbol{\alpha}_K(\boldsymbol{\theta}_t) \rangle \\ \text{s.t.} & \boldsymbol{\alpha} \in \mathcal{A}, \|\boldsymbol{\alpha} - \boldsymbol{\alpha}_K(\boldsymbol{\theta}_t)\| \leq 1 \end{array}$$

$$= -\langle \nabla_{\boldsymbol{\alpha}} f_\lambda(\boldsymbol{\theta}_t, \boldsymbol{\alpha}^*(\boldsymbol{\theta}_t)), \boldsymbol{\alpha}^*(\boldsymbol{\theta}_t) - \boldsymbol{\alpha}_K(\boldsymbol{\theta}_t) \rangle$$
$$- \underbrace{\begin{array}{ll} \max_{\boldsymbol{\alpha}} & \langle \nabla_{\boldsymbol{\alpha}} f_\lambda(\boldsymbol{\theta}_t, \boldsymbol{\alpha}^*(\boldsymbol{\theta}_t)), \boldsymbol{\alpha} - \boldsymbol{\alpha}^*(\boldsymbol{\theta}_t) \rangle \\ \text{s.t.} & \boldsymbol{\alpha} \in \mathcal{A}, \|\boldsymbol{\alpha} - \boldsymbol{\alpha}_K(\boldsymbol{\theta}_t)\| \leq 1 \end{array}}_{0},$$

$$= -\langle \nabla_{\boldsymbol{\alpha}} f_\lambda(\boldsymbol{\theta}_t, \boldsymbol{\alpha}^*(\boldsymbol{\theta}_t)), \boldsymbol{\alpha}^*(\boldsymbol{\theta}_t) - \boldsymbol{\alpha}_K(\boldsymbol{\theta}_t) \rangle$$

(50)

where the last equality holds since $\boldsymbol{\alpha}^*(\boldsymbol{\theta}_t)$ is optimal and $\|\boldsymbol{\alpha}^*(\boldsymbol{\theta}_t) - \boldsymbol{\alpha}_K(\boldsymbol{\theta}_t)\| \leq 1$. Combining (49) and (50), we get

$$\begin{array}{ll} \min_s & -\langle \nabla_{\boldsymbol{\alpha}} f_\lambda(\boldsymbol{\theta}_t, \boldsymbol{\alpha}_K(\boldsymbol{\theta}_t)), s \rangle \\ \text{s.t.} & \boldsymbol{\alpha}_K(\boldsymbol{\theta}_t) + s \in \mathcal{A}, \|s\| \leq 1 \end{array} \geq -\big( \|\nabla_{\boldsymbol{\alpha}} f_\lambda(\boldsymbol{\theta}_t, \boldsymbol{\alpha}^*(\boldsymbol{\theta}_t))\| + 2L_{22} \big) \|\boldsymbol{\alpha}_K(\boldsymbol{\theta}_t) - \boldsymbol{\alpha}^*\|.$$

(51)

Hence, using (11), we get

$$\mathcal{Y}_{t,K} \leq \big( 2L_{22} + g_{max} \big) \|\boldsymbol{\alpha}_K(\boldsymbol{\theta}_t) - \boldsymbol{\alpha}^*\|$$

$$\leq \frac{3\bar{L}}{2^{K/2\sqrt{8\kappa}}} \sqrt{\frac{2\Delta}{\lambda}} \tag{52}$$

$$\leq \frac{\varepsilon}{2},$$

where the second inequality uses (47), and the last inequality holds by our choice of $K$ in (43) and since $\varepsilon \in (0, 1)$. $\qquad\square$

The above lemma implies that $\|\nabla_{\boldsymbol{\theta}} f_\lambda(\boldsymbol{\theta}_t, \boldsymbol{\alpha}_K(\boldsymbol{\theta}_t)) - \nabla g_\lambda(\boldsymbol{\theta}_t)\| \leq \delta \triangleq \frac{L\varepsilon^2}{64\bar{R}^3\bar{L}^2}$. We now show that our assumption $g(\boldsymbol{\theta}_t) - f(\boldsymbol{\theta}_t, \boldsymbol{\alpha}_0(\boldsymbol{\theta}_t)) \leq \Delta$ for all t in the above Lemma holds. Let

$$\boldsymbol{\alpha}^*_{t+1} \triangleq \arg\max_{\boldsymbol{\alpha} \in \mathcal{A}} f_\lambda(\boldsymbol{\theta}_{t+1}, \boldsymbol{\alpha}) \quad \text{and} \quad \boldsymbol{\alpha}^*_t \triangleq \arg\max_{\boldsymbol{\alpha} \in \mathcal{A}} f_\lambda(\boldsymbol{\theta}_t, \boldsymbol{\alpha}).$$

Then by strong convexity of $-f_\lambda(\boldsymbol{\theta}, \cdot)$, we have

$$f_\lambda(\boldsymbol{\theta}_{t+1}, \boldsymbol{\alpha}^*_{t+1}) \leq f_\lambda(\boldsymbol{\theta}_{t+1}, \boldsymbol{\alpha}^*_t) + \langle \nabla_{\boldsymbol{\alpha}} f_\lambda(\boldsymbol{\theta}_{t+1}, \boldsymbol{\alpha}^*_t), \boldsymbol{\alpha}^*_{t+1} - \boldsymbol{\alpha}^*_t \rangle - \frac{\lambda}{2} \|\boldsymbol{\alpha}^*_{t+1} - \boldsymbol{\alpha}^*_t\|^2,$$

and

$$f_\lambda(\boldsymbol{\theta}_{t+1}, \boldsymbol{\alpha}^*_t) \leq f_\lambda(\boldsymbol{\theta}_{t+1}, \boldsymbol{\alpha}^*_{t+1}) + \underbrace{\langle \nabla_{\boldsymbol{\alpha}} f_\lambda(\boldsymbol{\theta}_{t+1}, \boldsymbol{\alpha}^*_{t+1}), \boldsymbol{\alpha}^*_t - \boldsymbol{\alpha}^*_{t+1} \rangle}_{\leq 0, \text{ by optimality of } \boldsymbol{\alpha}^*_{t+1}} - \frac{\lambda}{2} \|\boldsymbol{\alpha}^*_{t+1} - \boldsymbol{\alpha}^*_t\|^2.$$

Adding the two inequalities, we get

$$\langle \nabla_{\boldsymbol{\alpha}} f_\lambda(\boldsymbol{\theta}_{t+1}, \boldsymbol{\alpha}^*_t), \boldsymbol{\alpha}^*_{t+1} - \boldsymbol{\alpha}^*_t \rangle \geq \lambda \|\boldsymbol{\alpha}^*_{t+1} - \boldsymbol{\alpha}^*_t\|^2. \tag{53}$$

Moreover, due to optimality of $\boldsymbol{\alpha}_t^*$, we have

$$\langle \nabla_{\boldsymbol{\alpha}} f_\lambda(\boldsymbol{\theta}_t, \boldsymbol{\alpha}_t^*), \boldsymbol{\alpha}_{t+1}^* - \boldsymbol{\alpha}_t^* \rangle \le 0. \tag{54}$$

Combining (53) and (54) we obtain

$$
\begin{aligned}
\lambda \|\boldsymbol{\alpha}_{t+1}^* - \boldsymbol{\alpha}_t^*\|^2 & \le \langle \nabla_{\boldsymbol{\alpha}} f_\lambda(\boldsymbol{\theta}_{t+1}, \boldsymbol{\alpha}_t^*) - \nabla_{\boldsymbol{\alpha}} f_\lambda(\boldsymbol{\theta}_t, \boldsymbol{\alpha}_t^*), \boldsymbol{\alpha}_{t+1}^* - \boldsymbol{\alpha}_t^* \rangle \\
& \le L_{12} \|\boldsymbol{\theta}_t - \boldsymbol{\theta}_{t+1}\| \|\boldsymbol{\alpha}_{t+1}^* - \boldsymbol{\alpha}_t^*\|,
\end{aligned}
\tag{55}
$$

Thus,

$$\|\boldsymbol{\alpha}_{t+1} - \boldsymbol{\alpha}_t\| \le \frac{L_{12}}{\lambda} \|\boldsymbol{\theta}_{t+1} - \boldsymbol{\theta}_t\|.$$

Hence, the difference between consecutive optimal solutions computed by the inner loop of the algorithm, are upper bounded by the difference between corresponding $\boldsymbol{\theta}$'s. Since $\Theta$ is a compact set, we can find an upper bound $\Delta$ such that $g(\boldsymbol{\theta}_t) - f(\boldsymbol{\theta}_t, \alpha_0(\boldsymbol{\theta}_t)) \le \Delta$, for all $t$.

We are now ready to show the main theorem that implies convergence of our proposed algorithm to an $\varepsilon$–first-order stationary solution of problem (2). In particular, we show that using $\nabla_{\boldsymbol{\theta}} f_\lambda(\boldsymbol{\theta}_t, \boldsymbol{\alpha}_K(\boldsymbol{\theta}_t))$ instead of $\nabla g_\lambda(\boldsymbol{\theta}_t)$ for a small enough $\lambda$ in the Frank-Wolfe or projected descent algorithms applied to $g_\lambda$, finds an $\varepsilon$–FNE. We are now ready to show Theorem 4.2.

*Proof.* **Frank-Wolfe Steps:** We now show the result when Step 7 of Algorithm 2 sets

$$\boldsymbol{\theta}_{t+1} = \boldsymbol{\theta}_t + \frac{\mathcal{X}_t}{\widetilde{L}} \widehat{s}_t.$$

Using descent lemma on $g_\lambda$ and the definition of $\widetilde{L}$ in Algorithm 2, we have

$$
\begin{aligned}
g_\lambda(\boldsymbol{\theta}_{t+1}) & \le g_\lambda(\boldsymbol{\theta}_t) + \langle \nabla g_\lambda(\boldsymbol{\theta}_t), \boldsymbol{\theta}_{t+1} - \boldsymbol{\theta}_t \rangle + \frac{\widetilde{L}}{2} \|\boldsymbol{\theta}_{t+1} - \boldsymbol{\theta}_t\|^2 \\[2mm]
& = g_\lambda(\boldsymbol{\theta}_t) + \frac{\mathcal{X}_t}{\widetilde{L}} \langle \nabla g_\lambda(\boldsymbol{\theta}_t), \widehat{s}_t \rangle + \frac{\mathcal{X}_t^2}{2\widetilde{L}} \|\widehat{s}_t\|^2 \\[2mm]
& \le g_\lambda(\boldsymbol{\theta}_t) + \frac{\mathcal{X}_t}{\widetilde{L}} \langle \nabla g_\lambda(\boldsymbol{\theta}_t), \widehat{s}_t \rangle + \frac{\mathcal{X}_t^2}{2\widetilde{L}} \\[2mm]
& = g_\lambda(\boldsymbol{\theta}_t) - \frac{\mathcal{X}_t}{\widetilde{L}} \langle \underbrace{\nabla_{\boldsymbol{\theta}} f_\lambda(\boldsymbol{\theta}_t, \boldsymbol{\alpha}_K(\boldsymbol{\theta}_t)) - \nabla g_\lambda(\boldsymbol{\theta}_t)}_{e_t}, \widehat{s}_t \rangle - \frac{\mathcal{X}_t^2}{2\widetilde{L}} \\[2mm]
& \le g_\lambda(\boldsymbol{\theta}_t) + \frac{\mathcal{X}_t}{\widetilde{L}} \|e_t\| - \frac{\mathcal{X}_t^2}{2\widetilde{L}}
\end{aligned}
\tag{56}
$$

where $\widehat{s}_t$ and $\mathcal{X}_t$ are defined in equations (35) and (36) of the manuscript, and the second and last inequalities use the fact that $\|\widehat{s}_t\| \le 1$.

Summing up these inequalities for all values of $t$ leads to

$$\frac{1}{T} \sum_{t=0}^{T-1} \mathcal{X}_t^2 \le \frac{2\widetilde{L}\Delta}{T} + 4\|e_t\| g_{max} \le \frac{2\widetilde{L}\Delta}{T} + \frac{\varepsilon^2}{4} \le \frac{\varepsilon^2}{2}, \tag{57}$$

where the first inequality holds since

$$
\begin{aligned}
\mathcal{X}_t & = \langle \nabla_{\boldsymbol{\theta}} f_\lambda(\boldsymbol{\theta}_t, \boldsymbol{\alpha}_K(\boldsymbol{\theta}_t)) - \nabla_{\boldsymbol{\theta}} f_\lambda(\boldsymbol{\theta}_t, \boldsymbol{\alpha}^*(\boldsymbol{\theta}_t)) + \nabla_{\boldsymbol{\theta}} f_\lambda(\boldsymbol{\theta}_t, \boldsymbol{\alpha}^*(\boldsymbol{\theta}_t)), \widehat{s}_t \rangle \\[2mm]
& \le g_{max} + \|e_t\| \\[2mm]
& \le 2g_{max}.
\end{aligned}
$$

Here the first inequality in (57) holds by (11), Cauchy-Schwartz, and the fact that $\|\widehat{s}_t\| \leq 1$. The last inequality holds by our choice of $K$ in Lemma B.3

$$K \geq N_K(\varepsilon) \triangleq \frac{\sqrt{8\kappa}}{\log 2}\left(4\log(1/\varepsilon) + \log(2^{17}\bar{L}^6\bar{R}^6\Delta/L^2\lambda)\right),$$

which yields $\|e_t\| \leq 1 \leq g_{max}$ and by choosing $T$ such that

$$T \geq N_T(\varepsilon) \triangleq \frac{8\widetilde{L}\Delta}{\varepsilon^2}.$$

Therefore, using Lemma B.3, there exists at least one index $\widehat{t}$ for which

$$\mathcal{X}_{\widehat{t}} \leq \varepsilon \quad \text{and} \quad \mathcal{Y}_{\widehat{t},K} \leq \frac{\varepsilon}{2}. \tag{58}$$

Hence,

$$
\begin{aligned}
\mathcal{Y}(\boldsymbol{\theta}_{\widehat{t}}, \boldsymbol{\alpha}_K(\boldsymbol{\theta}_{\widehat{t}})) &= \begin{array}{c} \max_s \langle \nabla_{\boldsymbol{\alpha}} f(\boldsymbol{\theta}_{\widehat{t}}, \boldsymbol{\alpha}_K(\boldsymbol{\theta}_{\widehat{t}})), s \rangle \\ \text{s.t.} \quad \boldsymbol{\alpha}_{\text{K}}(\boldsymbol{\theta}_{\widehat{t}}) + \text{s} \in \mathcal{A}, \|\text{s}\| \leq 1 \end{array} \\
&= \begin{array}{c} \max_s \langle \nabla_{\boldsymbol{\alpha}} f_\lambda(\boldsymbol{\theta}_{\widehat{t}}, \boldsymbol{\alpha}_K(\boldsymbol{\theta}_{\widehat{t}})), s \rangle + \lambda(\boldsymbol{\alpha}_K(\boldsymbol{\theta}_{\widehat{t}}) - \bar{\boldsymbol{\alpha}})^T s \\ \text{s.t.} \quad \boldsymbol{\alpha}_{\text{K}}(\boldsymbol{\theta}_{\widehat{t}}) + \text{s} \in \mathcal{A}, \|\text{s}\| \leq 1 \end{array} \\
&\leq \mathcal{Y}_{\widehat{t},K} + \lambda\|\boldsymbol{\alpha}_K(\boldsymbol{\theta}_{\widehat{t}}) - \bar{\boldsymbol{\alpha}}\| \\
&\leq \varepsilon
\end{aligned}
\tag{59}
$$

where the first inequality uses Cauchy Shwartz and the fact that $\|s\| \leq 1$, and the last inequality holds due to (58), the choice of $\lambda$ in the theorem and our assumption that $\|\boldsymbol{\alpha}_K(\boldsymbol{\theta}_{\widehat{t}}) - \bar{\boldsymbol{\alpha}}\| \leq 2R$.

**Projected Gradient Descent:**
We start by defining
$$\Delta_g = g_\lambda(\boldsymbol{\theta}_0) - g^*,$$
where $g_\lambda^* \triangleq \min_{\boldsymbol{\theta}} g_\lambda(\boldsymbol{\theta})$ is the optimal value of $g_\lambda$. Note that by the compactness assumption of the set $\Theta$, we have $\Delta_g = g_\lambda(\theta_0) - g_\lambda^* < \infty$.

We now show the result when Step 7 of Algorithm 2 sets

$$\boldsymbol{\theta}_{t+1} = \text{proj}_\Theta\left(\boldsymbol{\theta}_{\text{t}} - \frac{1}{\text{L}}\nabla_{\boldsymbol{\theta}} f_\lambda(\boldsymbol{\theta}_{\text{t}}, \boldsymbol{\alpha}_{\text{K}}(\text{t}))\right),$$

Based on the projection property, we know that

$$\left\langle \boldsymbol{\theta}_t - \frac{1}{L}\nabla_{\boldsymbol{\theta}} f(\boldsymbol{\theta}_t, \boldsymbol{\alpha}_{t+1}) - \boldsymbol{\theta}_{t+1}, \boldsymbol{\theta} - \boldsymbol{\theta}_{t+1}\right\rangle \leq 0 \quad \forall\ \boldsymbol{\theta} \in \Theta.$$

Therefore, by setting $\boldsymbol{\theta} = \boldsymbol{\theta}_t$, we get

$$\left\langle \nabla_{\boldsymbol{\theta}} f(\boldsymbol{\theta}_t, \boldsymbol{\alpha}_{t+1}), \boldsymbol{\theta}_{t+1} - \boldsymbol{\theta}_t\right\rangle \leq -L\|\boldsymbol{\theta}_t - \boldsymbol{\theta}_{t+1}\|^2,$$

which implies

$$\left\langle \nabla_{\boldsymbol{\theta}} f\left(\boldsymbol{\theta}_t, \boldsymbol{\alpha}^*(\boldsymbol{\theta}_t)\right), \boldsymbol{\theta}_{t+1} - \boldsymbol{\theta}_t\right\rangle \leq -L\|\boldsymbol{\theta}_t - \boldsymbol{\theta}_{t+1}\|^2 + \left\langle \nabla_{\boldsymbol{\theta}} f\left(\boldsymbol{\theta}_t, \boldsymbol{\alpha}^*(\boldsymbol{\theta}_t)\right) - \nabla_{\boldsymbol{\theta}} f\left(\boldsymbol{\theta}_t, \boldsymbol{\alpha}_{t+1}\right), \boldsymbol{\theta}_{t+1} - \boldsymbol{\theta}_t\right\rangle$$

$$= -L\|\boldsymbol{\theta}_t - \boldsymbol{\theta}_{t+1}\|^2 + \left\langle e_t, \boldsymbol{\theta}_t - \boldsymbol{\theta}_{t+1}\right\rangle$$

$$\tag{60}$$

where $\boldsymbol{\alpha}^*(\boldsymbol{\theta}_t) \triangleq \arg\max_{\boldsymbol{\alpha}\in\mathcal{A}} f_\lambda(\boldsymbol{\theta}_t, \boldsymbol{\alpha})$ and $e_t \triangleq \nabla_{\boldsymbol{\theta}} f\left(\boldsymbol{\theta}_t, \boldsymbol{\alpha}_{t+1}\right) - \nabla_{\boldsymbol{\theta}} f\left(\boldsymbol{\theta}_t, \boldsymbol{\alpha}^*(\boldsymbol{\theta}_t)\right)$.

By Taylor expansion, we have

$$g_\lambda(\boldsymbol{\theta}_{t+1}) \le g_\lambda(\boldsymbol{\theta}_t) + \left\langle \nabla_{\boldsymbol{\theta}} f\big(\boldsymbol{\theta}_t, \boldsymbol{\alpha}^*(\boldsymbol{\theta}_t)\big), \boldsymbol{\theta}_{t+1} - \boldsymbol{\theta}_t \right\rangle + \frac{L}{2}\|\boldsymbol{\theta}_{t+1} - \boldsymbol{\theta}_t\|^2$$

$$\tag{61}$$

$$\le g_\lambda(\boldsymbol{\theta}_t) - \frac{L}{2}\|\boldsymbol{\theta}_{t+1} - \boldsymbol{\theta}_t\|^2 + \langle e_t, \boldsymbol{\theta}_t - \boldsymbol{\theta}_{t+1}\rangle.$$

Moreover, by the projection property, we know that

$$\left\langle \nabla_{\boldsymbol{\theta}} f(\boldsymbol{\theta}_t, \boldsymbol{\alpha}_{t+1}), \boldsymbol{\theta} - \boldsymbol{\theta}_{t+1}\right\rangle \ge L\langle \boldsymbol{\theta}_t - \boldsymbol{\theta}_{t+1}, \boldsymbol{\theta} - \boldsymbol{\theta}_{t+1}\rangle,$$

which implies

$$\left\langle \nabla_{\boldsymbol{\theta}} f(\boldsymbol{\theta}_t, \boldsymbol{\alpha}_{t+1}), \boldsymbol{\theta} - \boldsymbol{\theta}_t\right\rangle \ge \left\langle \nabla_{\boldsymbol{\theta}} f(\boldsymbol{\theta}_t, \boldsymbol{\alpha}_{t+1}), \boldsymbol{\theta}_{t+1} - \boldsymbol{\theta}_t\right\rangle + L\langle \boldsymbol{\theta}_t - \boldsymbol{\theta}_{t+1}, \boldsymbol{\theta} - \boldsymbol{\theta}_{t+1}\rangle$$

$$\ge -(g_{max} + 2LR + \|e_t\|)\|\boldsymbol{\theta}_{t+1} - \boldsymbol{\theta}_t\| \tag{62}$$

$$\ge -2(g_{max} + LR)\|\boldsymbol{\theta}_{t+1} - \boldsymbol{\theta}_t\|.$$

Here the second inequality holds by Cauchy-Schwartz, the definition of $e_t$ and our assumption that $\Theta \subseteq \mathcal{B}_R$. Moreover, the last inequality holds by our choice of $K$ in Lemma A.6 which yields

$$\|e_t\| = \|\nabla_{\boldsymbol{\theta}} f(\boldsymbol{\theta}_t, \boldsymbol{\alpha}_K(\boldsymbol{\theta}_t)) - \nabla g(\boldsymbol{\theta})\| \tag{63}$$
$$\le L_{12}\|\boldsymbol{\alpha}_K(\boldsymbol{\theta}_t) - \boldsymbol{\alpha}^*\|$$

$$\le L_{12}\rho^{K/2}\sqrt{\frac{\Delta}{2\mu}}$$

$$\le 1 \tag{64}$$
$$\le g_{max}. \tag{65}$$

Hence,

$$-\mathcal{X}_t \ge -2(g_{max} + LR)\|\boldsymbol{\theta}_{t+1} - \boldsymbol{\theta}_t\|,$$

or equivalently

$$\|\boldsymbol{\theta}_{t+1} - \boldsymbol{\theta}_t\| \ge \frac{\mathcal{X}_t}{2(g_{max} + LR)}. \tag{66}$$

Combined with (61), we get

$$g_\lambda(\boldsymbol{\theta}_{t+1}) - g_\lambda(\boldsymbol{\theta}_t) \quad \le -\frac{L}{8}\frac{\mathcal{X}_t^2}{\big(g_{max} + LR\big)^2} + 2\|e_t\|R,$$

where the inequality holds by using Cauchy Schwartz and our assumption that $\Theta$ is in a ball of radius $R$. Hence,

$$\frac{1}{T}\sum_{t=0}^{T-1} \mathcal{X}_t^2 \le \frac{8\Delta_g(g_{max} + LR)^2}{LT} + \frac{16\delta R(g_{max} + LR)^2}{L}$$

$$\le \frac{\varepsilon^2}{2},$$

where the last inequality holds by using Lemma B.3 and choosing $K$ and $T$:

$$T \ge N_T(\varepsilon) \triangleq \frac{32\Delta_g(g_{max} + LR)^2}{L\varepsilon^2}, \quad \text{and} \quad K \ge N_K(\varepsilon) \triangleq \frac{\sqrt{8\kappa}}{\log 2}\big(4\log(1/\varepsilon) + \log(2^{17}\bar{L}^6 \bar{R}^6 \Delta/L^2\lambda)\big),$$

Therefore, using Lemma B.3, there exists at least one index $\widehat{t}$ for which

$$\mathcal{X}_{\widehat{t}} \le \varepsilon \quad \text{and} \quad \mathcal{Y}_{\widehat{t},K} \le \frac{\varepsilon}{2}. \tag{67}$$

Hence,

$$
\begin{aligned}
\mathcal{Y}(\boldsymbol{\theta}_{\widehat{t}}, \boldsymbol{\alpha}_K(\boldsymbol{\theta}_{\widehat{t}})) \;=\; & \max_s \; \langle \nabla_{\boldsymbol{\alpha}} f(\boldsymbol{\theta}_{\widehat{t}}, \boldsymbol{\alpha}_K(\boldsymbol{\theta}_{\widehat{t}})), s \rangle \\
& \text{s.t.} \quad \boldsymbol{\alpha}_K(\boldsymbol{\theta}_{\widehat{t}}) + s \in \mathcal{A}, \; \|s\| \le 1 \\[4pt]
\;=\; & \max_s \; \langle \nabla_{\boldsymbol{\alpha}} f_\lambda(\boldsymbol{\theta}_{\widehat{t}}, \boldsymbol{\alpha}_K(\boldsymbol{\theta}_{\widehat{t}})), s \rangle + \lambda(\boldsymbol{\alpha}_K(\boldsymbol{\theta}_{\widehat{t}}) - \bar{\boldsymbol{\alpha}})^T s \\
& \text{s.t.} \quad \boldsymbol{\alpha}_K(\boldsymbol{\theta}_{\widehat{t}}) + s \in \mathcal{A}, \; \|s\| \le 1 \\[4pt]
\;\le\; & \mathcal{Y}_{\widehat{t},K} + \lambda \|\boldsymbol{\alpha}_K(\boldsymbol{\theta}_{\widehat{t}}) - \bar{\boldsymbol{\alpha}}\| \\[4pt]
\;\le\; & \varepsilon
\end{aligned}
\tag{68}
$$

where the first inequality uses Cauchy Shwartz and the fact that $\|s\| \le 1$, and the last inequality holds due to (67), the choice of $\lambda$ in the theorem and our assumption that $\|\boldsymbol{\alpha}_K(\boldsymbol{\theta}_{\widehat{t}}) - \bar{\boldsymbol{\alpha}}\| \le 2R$.

$\square$

## C  Numerical Results on Fashion MNIST with SGD

The results of using SGD optimizer are summarized in Table 4 and Table 5. Note SGD optimizer requires more tuning and therefore the results when batch-size = 3000 is also included here.

| | T-shirt/top | | Coat | | Shirt | | Worst | |
|---|---|---|---|---|---|---|---|---|
| | mean | std | mean | std | mean | std | mean | std |
| Normal | 850.26 | 8.59 | 806.78 | 18.92 | 558.72 | 30.99 | 558.72 | 30.99 |
| MinMax | 754.68 | 12.03 | 699.04 | 28.76 | 724.86 | 18.00 | 696.60 | 25.93 |
| MinMax with Regularization | 756.16 | 13.60 | 701.02 | 30.07 | 723.14 | 18.52 | 698.16 | 26.96 |

Table 4: The mean and standard deviation of the number of correctly classified samples when SGD (mini-batch) is used in training, $\lambda = 0.05$, batch-size = 3000.

| | T-shirt/top | | Coat | | Shirt | | Worst | |
|---|---|---|---|---|---|---|---|---|
| | mean | std | mean | std | mean | std | mean | std |
| Normal | 849.76 | 8.20 | 807.60 | 19.19 | 563.90 | 29.64 | 563.90 | 29.64 |
| MinMax | 755.34 | 13.72 | 702.60 | 26.11 | 723.70 | 18.92 | 700.46 | 24.02 |
| MinMax with Regularization | 754.78 | 14.92 | 703.70 | 24.80 | 723.44 | 19.29 | 701.78 | 23.13 |

Table 5: The mean and standard deviation of the number of correctly classified samples when SGD (mini-batch) is used in training, $\lambda = 0.0005$, batch-size = 600.

## D  Numerical Results on Fashion MNIST with Logistic Rgression Model

Table 6 shows that the proposed formulation gives better accuracies under the worst category (Shirts), and the accuracies over three categories are more balanced. Note that this model is trained by gradient descent. The standard derivations not equal to 0 is due to the early termination of the simulation.

|  | T-shirt/top | | Pullover | | Shirt | |
|---|---|---|---|---|---|---|
|  | mean | std | mean | std | mean | std |
| [42] | 849.00 | 44.00 | 876.00 | 45.00 | 745.00 | 60.00 |
| Proposed | 778.48 | 8.78 | 773.46 | 8.76 | 740.60 | 9.26 |

Table 6: The mean and standard deviation of the number of correctly classified samples when gradient descent is used in training, $\lambda = 0.1$.

# E  Numerical Results on Robust Neural Network Training

Neural networks have been widely used in various applications, especially in the field of image recognition. However, these neural networks are vulnerable to adversarial attacks, such as Fast Gradient Sign Method (FGSM) [25] and Projected Gradient Descent (PGD) attack [31]. These adversarial attacks show that a small perturbation in the data input can significantly change the output of a neural network. To train a robust neural network against adversarial attacks, researchers reformulate the training procedure into a robust min-max optimization formulation [38], such as

$$\min_{\mathbf{w}} \sum_{i=1}^{N} \max_{\delta_i, \text{ s.t. } |\delta_i|_\infty \leq \varepsilon} \ell(f(x_i + \delta_i; \mathbf{w}), y_i).$$

Here $\mathbf{w}$ is the parameter of the neural network, the pair $(x_i, y_i)$ denotes the $i$-th data point, and $\delta_i$ is the perturbation added to data point $i$. As discussed in this paper, solving such a non-convex non-concave min-max optimization problem is computationally challenging. Motivated by the theory developed in this work, we approximate the above optimization problem with a novel min-max objective function which has concave inner optimization problem. To do so, we first approximate the inner maximization problem with a finite max problem

$$\min_{\mathbf{w}} \sum_{i=1}^{N} \max \left\{ \ell(f(\hat{x}_{i0}(\mathbf{w}); \mathbf{w}), y_i), \ldots, \ell(f(\hat{x}_{i9}(\mathbf{w}); \mathbf{w}), y_i) \right\}, \tag{69}$$

where each $\hat{x}_{ij}(\mathbf{w})$ is the result of a targeted attack on sample $x_i$ aiming at changing the output of the network to label $j$. More specifically, $\hat{x}_{ij}(\mathbf{w})$ is obtained through the following procedure:

In the one but last layer of the neural network architecture for learning classification on MNIST we have 10 different neurons, each corresponding with one category of classification. For any sample $(x_i, y_i)$ in the dataset and any $j = 0, \cdots, 9$, starting from $x_{ij}^0 = x_i$, we run gradient ascent to obtain the following chain of points:

$$x_{ij}^{k+1} = \text{Proj}_{B(x,\varepsilon)} \left[ x_{ij}^k + \alpha \nabla_x (Z_j(x_{ij}^k, \mathbf{w}) - Z_{y_i}(x_{ij}^k, \mathbf{w})) \right], \ k = 0, \cdots, K-1,$$

where $Z_j$ is the network logit before softmax corresponding to label $j$; $\alpha > 0$ is the step-size; and $\text{Proj}_{B(x,\varepsilon)}[\cdot]$ is the projection to the infinity ball with radius $\varepsilon$ centered at $x$. Finally, we set $\hat{x}_{ij}(\mathbf{w}) = x_{ij}^K$ in (69).

Clearly, we can replace the finite max problem (69) with a concave problem over a probability simplex, i.e.,

$$\min_{\mathbf{w}} \sum_{i=1}^{N} \max_{\mathbf{t} \in \mathcal{T}} \sum_{j=0}^{9} t_j \ell\left( f\left(x_{ij}^K; \mathbf{w}\right), y_i \right), \ \mathcal{T} = \{\mathbf{t} \in \mathbb{R}^{10} \, | \, \mathbf{t} \geq 0, \ ||\mathbf{t}||_1 = 1\}, \tag{70}$$

which is non-convex in $w$, but concave in $\mathbf{t}$. Hence we can apply Algorithm 2 to solve this opimization problem. We test (70) on MNIST dataset with a Convolutional Neural Network(CNN) with the architecture detailed in Table 7. The result of our experiment is presented in Table 8.

| Layer Type | Shape |
|---|---|
| Convolution + ReLU | $5 \times 5 \times 20$ |
| Max Pooling | $2 \times 2$ |
| Convolution + ReLU | $5 \times 5 \times 50$ |
| Max Pooling | $2 \times 2$ |
| Fully Connected + ReLU | 800 |
| Fully Connected + ReLU | 500 |
| Softmax | 10 |

Table 7: Model Architecture for the MNIST dataset.

| | Natural | FGSM $L_\infty$ [25] | | | PGD$^{40}$ $L_\infty$ [31] | | |
|---|---|---|---|---|---|---|---|
| | | $\varepsilon = 0.2$ | $\varepsilon = 0.3$ | $\varepsilon = 0.4$ | $\varepsilon = 0.2$ | $\varepsilon = 0.3$ | $\varepsilon = 0.4$ |
| [38] with $\varepsilon = 0.35$ | **98.58%** | 96.09% | 94.82% | 89.84% | 94.64% | 91.41% | 78.67% |
| [57] with $\varepsilon = 0.35$ | 97.37% | 95.47% | 94.86% | 79.04% | 94.41% | 92.69% | 85.74% |
| [57] with $\varepsilon = 0.40$ | 97.21% | 96.19% | 96.17% | 96.14% | 95.01% | 94.36% | 94.11% |
| Proposed with $\varepsilon = 0.40$ | 98.20% | **97.04%** | **96.66%** | **96.23%** | **96.00%** | **95.17%** | **94.22%** |

Table 8: Test accuracies under FGSM and PGD attacks. We set $K = 10$ to train our model, and we take step-size 0.01 when generating PGD attacks. All adversarial images are quantified to 256 levels $(0 - 255$ integer).

**Remark E.1.** *We would like to note that there is a mismatch between our theory and this numerical experiment. In particular, we assume smoothness of the objective function in our theory. However, in this experiment, the ReLu activation functions and the projection operator make the objective function non-smooth. We also did not include regularizer (strongly concave term) while solving* (70) *as the optimal regularizer was very small (and almost zero).*

**Remark E.2.** *The main take away from this experiment is to demonstrate the practicality of the following idea: when solving general challenging non-convex min-max problems, it might be possible to approximate it with one-sided non-convex min-max problems where the objective function is solvable with respect to one of the player's variable. Such a reformulation leads to computationally tractable problems and (possibly) no loss in the performance.*

# F   Experimental Setup of Fair Classifier

| Layer Type | Shape |
|---|---|
| Convolution + tanh | $3 \times 3 \times 5$ |
| Max Pooling | $2 \times 2$ |
| Convolution + tanh | $3 \times 3 \times 10$ |
| Max Pooling | $2 \times 2$ |
| Fully Connected + tanh | 250 |
| Fully Connected + tanh | 100 |
| Softmax | 3 |

Table 9: Model Architecture for the Fashion MNIST dataset. [55]

| Parameter | | | |
|---|---|---|---|
| Learning Rate | 0.1 | 0.05 | 0.01 |
| Epochs | 4000 | 1000 | 500 |

Table 10: Training Parameters for the Fashion MNIST dataset with gradient descent. [55]

| Parameter | | | |
|---|---|---|---|
| Learning Rate | $10^{-4}$ | $10^{-5}$ | $10^{-6}$ |
| Iterations | 4000 | 4000 | 4000 |
| Batch-size | 600 | | |

Table 11: Training Parameters for the Fashion MNIST dataset with Adam. [55]

| Parameter | | | |
|---|---|---|---|
| Learning Rate | $10^{-3}$ | $10^{-4}$ | $10^{-5}$ |
| Iterations | 8000 | 8000 | 8000 |

Table 12: Training Parameters for the Fashion MNIST dataset with SGD. [55]

# G   Links

Robust NN Training: `https://github.com/optimization-for-data-driven-science/Robust-NN-Training`

Fair Classifier: `https://github.com/optimization-for-data-driven-science/FairFashionMNIST`