[Reviews · NeurIPS 2019]

Reviewer 1



In this paper, the convergence of the gradient descent-ascent algorithm is studied for solving min-max optimization problems, including nonconvex PL games and nonconvex-concave games. In general, the paper is well written. My comments are listed as below. 1) Table 1 should be better presented by elaborating on the differences between this work and the existing ones in the problem setup of (1), e.g., convexity, and unconstrained sets (namely, $\mathcal \Theta$ and $\mathcal A$). 2) I am not convinced about Remark 3.8: "one can easily extend the result of Theorem 3.4 to the stochastic setting". Will the stochastic gradient apply to both inner maximization and outer minimization steps? I assumed that the proposed analysis only applies to SGD, right? Will the analysis apply to the variant using Adam? If no, why was Adam used in Table 3? If yes, please elaborate on it. 3) In (4) and (5), is the radius $1$ chosen for ease of analysis? #################Post-feedback################## My questions have been answered in the rebuttal. Please be sure to make additional clarification and experiments in the final version.

Reviewer 2



Originality. The paper seems to be a simple extension of known results for non-convex optimization to saddle-point problems. I think, it is obvious that since the saddle-point problem is concave or restricted strongly concave in the variable w.r.t. which we maximize, we can use some method to find the approximate gradient w.r.t. variable in which we minimize. Then a gradient method with inexact oracle, e.g. http://papers.nips.cc/paper/6565-learning-supervised-pagerank-with-gradient-based-and-gradient-free-optimization-methods, can be applied to find an approximate stationary point. It is quite strange that the classical paper https://epubs.siam.org/doi/pdf/10.1137/S1052623403425629 is not cited in line 31. Also it seems that the general algorithm https://link.springer.com/article/10.1007/s10589-014-9673-9 can be applied to the setting considered in this submission. Probably, the complexity would be the same. Quality. As far as I see, the proof of Lemma A.5 is not complete. To use the Taylor expansion for g(.) one first needs to prove that it is a smooth function. Moreover, the expansion is used up to the second-order term and more smoothness is needed. Thus, the contribution for the PL-games is questionable. Other proofs seem to be correct. Nevertheless, it would be nice to add some more details on how the first inequality in line after line 530 in the supplementary is obtained. The same is for line after line 449 in the supplementary. In the experiments it would be nice to compare the results with the results of [37]. Does CNN have advantage in comparison with convex classifier? Clarity. Except the above things the paper is well written and easy to follow. Significance. Despite the optimization contribution seems to be marginal, the algorithm can be interesting to practitioners in the ML community. =============After rebuttal============= The authors have mostly resolved my concerns by their rebuttal. I still think that the paper is borderline in terms of the contribution. Not much is new made in terms of the optimization, but the resulting algorithms can be interesting to the ML community. I'm increasing my score.

Reviewer 3



The basic idea behind their algorithms is that of a projected gradient descent for the min player and inside the loop, one performs accelerated or simple gradient ascent respectively for non-convex concave or non-convex- PL games. For the case of the non-convex concave games, due to non-smoothness implications, the authors introduce some regularization to make the function smooth. The techniques are quite standard, nevertheless the result is quite decent.

Reviewer 4



This is a good paper that studies an interesting problem. The paper is well-written with good literature review and exposition. The idea of using the PL condition and the smoothing approach are well-motivated from optimization, but the application to game play is interesting. The technical results and analysis seem solid. The experiments are rather limited, it would be interesting to verify the results on more varied examples.

[Author Response · NeurIPS 2019]

We thank the reviewers for their thoughtful comments and feedback. Below we respond to the reviewers' concerns.

**Novelty (R4&R5):** The convergence rate of $O(\epsilon^{-3.5})$ sets new achievable baselines in the literature for the correspond-
ing class of min-max games, compared to the previously best known rates in the literature; see Table 1 in the paper. In
addition, the convergence analysis of the PL min-max games is new. We acknowledge that our solutions are built upon
seemingly simple and intuitive techniques yet delivering better convergence results. It is worth noting that bringing
these blocks together carefully and in the right order is necessary to obtain the reported rates. We also had to extend
some existing techniques in non-obvious ways to establish these results. For example, Lemma A.5 (Danskin's theorem
for PL functions) is not previously reported in the literature and is developed from scratch in this paper. As noted by R4,
this result is novel and could be of independent interest outside the context of this paper.

**Lemma A.5 (R4&R6):** In the proof of Lemma A.5, on lines 429-430, we are using the Taylor expansion of $f(\cdot)$
up to the first order. Hence, we need Lipschitz smoothness of the $\nabla f$, i.e., existence and boundedness of $\nabla^2 f$. We
will further clarify this point in the updated version. Note that in this lemma (lines 431-436), we actually prove the
differentiability of the function $g(\cdot)$ and its Lipschitz smoothness. The result of this lemma is not obvious due to the
fact that the optimal solution set of the inner maximization is not necessarily a singleton in the PL case as opposed to
the strongly concave case. We believe one source of confusion for the reviewer is that we did not explicitly state that
the Taylor expansion in the proof is for function $f$. This led to the concern of the reviewer about the validity of Taylor
expansion for function $g$, while this Taylor expansion is in fact for function $f$.

**Experiments (R4&R6):** We plan to expand our experimental results in multiple directions. 1) We have already
performed experiments on robust training of neural networks based on a non-convex concave min-max formulation
and compared our method with the state-of-the-art algorithms in this setting, namely [Madry et al 2017] and [Zhang
et al 2019]. If accepted, we plan to include it in the final submission. 2) We also plan to add numerical comparisons
with [37] using logistic regression (which is a convex model) (suggested by R4). 3) We used CNNs in our experiments
as an example of more expressive non-convex models that are used in practice. In fact, we reported better worst case
performance (higher worst class accuracy and lower variance) compared to logistic regression (R4). Note that due to
the non-convexity of this setting, the algorithm of [37] would not have any convergence guarantee. However, we plan to
numerically apply it to this problem as a heuristic and compare against it.

**Remark 3.8 (R3):** We apologize for the confusion. By stochastic gradients we mean applying SGD on the outer loop
while assuming that the inner problem is solved by an oracle, i.e., it is easily solvable. For example, in our experiments
the inner maximization has analytical solution when one computes the loss on each class. As you have rightfully
mentioned, the convergence only applies to SGD. We reported the results for Adam in our experiments as it was more
robust to the choice of step-size and thus was tuned easily. We will include this discussion alongside the results for
SGD in the final version. Note that the use of SGD or Adam does not change the overall takeaways of the experiments.

**Radius 1 in equations 4 & 5 (R3):** Radius 1 is used for normalization. Note that the use of a finite radius in (4) and
(5) guarantees that the optimum value would be bounded. But choosing radius to be 1 assures that (4) and (5) can be
directly applied to the unconstrained case without any changes to the definitions, see lines 111-112.

**Comparison to Dang & Lan [A] (R4):** Thanks for bringing up this paper. The algorithm in [A] is developed for
generalized monotone variational inequities (GMVI), which is different than our setup. To understand the difference,
consider the case where $\mathcal{A}$ is singleton and $\Theta$ is unconstrained. In this case, the problem becomes an unconstrained
minimization and GMVI is equivalent to $\langle \nabla_\theta f(\theta, \alpha_0) - \nabla_\theta f(\theta^*, \alpha_0), \theta - \theta^* \rangle \geq 0$. In this case, GMVI is limited in
the sense that it does not cover general non-convex problems (unlike our setup). In general, neither our setup implies
[A], nor [A] covers our setup. We will include a detailed discussion on the results in [A] in our revision.

**PL examples (R6):** Note that PL condition appears in many practical convex and non-convex problems; see lines
71-74 in the paper. In fact, it is more general than strong convexity because it allows the existence of multiple optimal
solutions, e.g., if $f(\cdot)$ is strongly convex and $A$ is a linear mapping, $f(A(\cdot))$ is PL. Thus, any practical problem where
the inside max is concave, but satisfy such a form, e.g. high-dimensional linear or logistic regression, would be a PL
game. In the non-convex optimization though, the examples are more specific; see [15, 17, 48, 27] for some. Example
3.1, i.e., generative adversarial imitation learning for LQR, is a great example and we definitely plan to expand it
and explain it more explicitly. It is worth noting that using non-convex game formulations in learning applications is
relatively new. We believe more specific examples would be found as the field is moving forward very quickly.

**Minor comments:** We will expand Table 1 to include more details of all the algorithms (R3). We will add more details
on how we obtained the expressions in lines 449 and 530 (R4). We agree that we should have cited Nemirovski's paper
on convex-concave problems. Thanks for bringing it up. We will cite it appropriately in our final version (R4).

[A] C. D. Dang and G. Lan. On the convergence properties of non-euclidean extragradient methods for variational
inequalities with generalized monotone operators.

[Meta-Review · NeurIPS 2019]

Dear authors: your paper was evaluated carefully by the reviewers, and there was broad agreement, both before and after the rebuttal, that the paper has nice results that are worthy of publication. The multi-step procedure for finding an equilibrium is relatively straightforward, but the fact that the PL condition can be used to establish a decent rate of convergence is a nice result. However, please consider the reviewers' comments when finalizing your camera-ready document.